# Characteristics of long-track tropopause polar vortices

Matthew T. Bray[1] and Steven M. Cavallo[1]

[1]School of Meteorology, University of Oklahoma, 120 David L Boren Blvd., Norman, OK 73072, USA

**Correspondence:** Matthew Bray (matthewbray1@ou.edu)

**Abstract.** Tropopause polar vortices (TPVs) are closed circulations centered on the tropopause that form and predominately reside in high latitudes. Due to their attendant flow, TPVs have been shown to influence surface weather features, and thus, a greater understanding of the dynamics of these features may improve our ability to forecast impactful weather events. In this study, we focus on the subset of TPVs that have lifetimes of longer than two weeks (the ninety-fifth percentile of all TPV cases between 1979 and 2018); these long-lived vortices offer a unique opportunity to study the conditions under which TPVs strengthen and analyze patterns of vortex formation and movement. Using ERA-Interim data, along with TPV tracks derived from the same reanalysis, we investigate the formation, motion, and development of these long-lived vortices. We find that these TPVs are significantly stronger, occur more often in the summer, and tend to remain more poleward than an average TPV. Similarly, these TPVs are shown to form at higher latitudes than average. Long-lived TPVs form predominately by splitting from existing vortices, but a notable minority seem to generate via dynamic processes in the absence of pre-existing TPVs. These non-likely split genesis events are found to occur in select geographic regions, driven by Rossby wave growth and breaking. Seasonal variations emerge in the life cycles of long-lived vortices; notably, winter TPVs progress more equatorward and generally grow to stronger amplitudes. These long-lived TPVs also appear as likely as any TPV to exit the Arctic and move into the mid-latitudes, doing so via two primary pathways: through Canada or Siberia.

## 1 Introduction

Tropopause polar vortices (TPVs) are a well-studied feature of the upper troposphere and lower stratosphere in high-latitude regions (Hakim and Canavan, 2005; Cavallo and Hakim, 2009, 2010, 2013). TPVs are defined as closed material contours that form coherent circulations on the tropopause (Hakim, 2000; Hakim and Canavan, 2005). For mathematical purposes, TPVs are generally described by their potential vorticity (PV) characteristics (Hakim and Canavan, 2005; Cavallo and Hakim, 2009). In the present study, we will primarily examine TPVs as potential temperature anomalies on the dynamic tropopause (defined as the 2 PVU surface). This approach has the advantage of requiring maps at only one standard level, in contrast to isentropic plots of potential vorticity (Morgan and Nielson-Gammon, 1998). These vortices generally possess radii of several hundred to one thousand kilometers and may last for days to months in some cases (Hakim, 2000; Hakim and Canavan, 2005). Although TPVs may be cold-core and cyclonic or warm-core and anticyclonic, in this study, TPV will refer exclusively to cyclonic TPVs (i.e., minima of potential temperature on the dynamic tropopause), which have more established impacts on Arctic cyclones

and other weather events (Cavallo and Hakim, 2009, 2010). Further, while TPVs occur in both the Arctic and Antarctic, the current study will be limited to Arctic TPVs only.

For this analysis, any tropopause PV anomaly which forms north of 60°N will be considered a TPV. This is a slight relaxation from the definitions proposed in some previous studies (e.g., Hakim and Canavan, 2005; Cavallo and Hakim, 2010) but is important to account for long-lived vortices which form in the Arctic but may travel outside of the Arctic for long periods. Previous climatological studies of cyclonic PV anomalies near the tropopause or lower stratosphere (not necessarily using the TPV definition employed here) have found these vortices to be widespread across the Arctic, with especially high preference for regions over high terrain and near recurrent storm tracks (Hakim and Canavan, 2005; Kew et al., 2010; Cavallo and Hakim, 2009; Portmann et al., 2021). These previous studies have employed a variety of feature tracking algorithms, but for this study anomalies are tracked using TPVTrack (discussed further in Sect. 2), which could lead to slight variations in spatial climatologies (Szapiro and Cavallo, 2018).

Cavallo and Hakim (2012) suggested that dynamical processes may be generally responsible for the formation of TPVs, with many new anomalies splitting from previously existing vortices. One potential, though not yet fully explored, dynamical genesis mechanism for TPVs is Rossby wave breaking (RWB). RWB events, defined as an overturning or irreversible deformation of the PV field, have been shown to produce TPV-like cyclonic potential vorticity anomalies in some cases (McIntyre and Palmer, 1983; Appenzeller and Davies, 1992). On dynamic tropopause or isentropic maps, these RWBs take the visual form of so-called streamers or filaments of low-potential temperature or high PV (Appenzeller and Davies, 1992; Martius et al., 2007). Previous climatological studies of northern hemisphere RWB events have identified two main frequency maxima: one over eastern Siberia and the northern Pacific ocean and another over the northern Atlantic ocean stretching into northern Europe (Peters and Waugh, 1996; Wernli and Sprenger, 2007). Classification systems of RWB events have generally centered on describing the wave break as either cyclonically or anticyclonically sheared, along with determining if the wave break is predominately poleward or equatorward (Thorncroft et al., 1993; Gabriel and Peters, 2008). RWB events are known to produce cyclonic and anticyclonic PV anomalies (Röthlisberger et al., 2018; Martius et al., 2010). In particular, the genesis of PV features poleward of the jet (as TPVs are) has been linked to cyclonic RWB events (Portmann et al., 2021). To our knowledge, though, no study has explicitly evaluated the connection between RWB events and TPV formation.

Radiative process have been shown to be a primary driver of TPV intensification (Cavallo and Hakim, 2010, 2012, 2013). In particular, TPVs strengthen primarily via longwave cooling that is generated by enhanced moisture gradients across the tropopause and reinforced by cloud-top cooling from mid-level clouds (Cavallo and Hakim, 2013). Latent heating from deeper cloud layers in the middle troposphere works to destroy PV and weaken TPVs, as does shortwave radiation (Cavallo and Hakim, 2012, 2013). The motion of TPVs is dominated by advection and other dynamical processes, with the background flow being the primary determiner of TPV movement (Hakim and Canavan, 2005; Szapiro and Cavallo, 2018). Vortices tend to be broken down either by dynamic factors such as high shear from interactions with mid-latitude jets and reabsorption into the background flow or by physical factors such as diabatic destruction by latent heating (Cavallo and Hakim, 2009; Portmann et al., 2021).

TPVs have previously been connected with a variety of impacts across the earth system both within and outside of the polar regions; an understanding of TPVs is closely tied with the ability to forecast these events. At a basic level, TPVs act as an upper-level PV anomaly within the PV-thinking framework, serving as a bolster for surface cyclogenesis (Hoskins et al., 1985). In particular, this is true for Arctic cyclones, which commonly develop in tandem with one or more TPVs (Simmonds and Rudeva, 2012, 2014; Tao et al., 2017). Arctic cyclones, in turn, are important drivers of sea ice variability on short time

scales via ice breakup and transport. This connection has been demonstrated through case studies (e.g., Simmonds and Rudeva, 2012; Zhang et al., 2013) and climatological investigations (e.g., Simmonds and Keay, 2009; Screen et al., 2011; Wang et al., 2020). TPV effects have also been noted to extend outside of the polar regions. TPVs have been connected to mid-latitude cyclone development and to other significant mid-latitude weather events (Hakim et al., 1995; Morgan and Nielson-Gammon, 1998; Hakim, 2000). In particular, TPVs, like other PV anomalies, may contribute to the development of jet streaks upon

interacting with a jet (Pyle et al., 2004). These TPV-induced jet maxima can then play a role in weather events ranging from heavy rainfall to tornado outbreaks (Bray et al., 2021; Hakim, 2000). Similarly, TPVs have been linked with Rossby wave initiation events, which have the potential to influence surface weather downstream (Röthlisberger et al., 2018; Kew et al., 2010). Furthermore, TPVs have been well established to play a role in many cold air outbreak events (Shapiro et al., 1987; Papritz et al., 2019; Lillo et al., 2021; Biernat et al., 2021). These extra-Arctic effects may be uniquely relevant to the current

study, as long-lived TPVs could progress especially far into the mid-latitudes.

While prior work has explored the physical mechanisms of TPVs and established climatologies of their occurrence, many characteristics of TPVs still remain unknown. This includes, for example, a comprehensive account of where and how they form, how they move throughout their lifetime, and why some vortices remain coherent for much longer than others, even outside of the Arctic. In the present study, we will focus on the subset of TPVs that have especially long lifetimes. Based

on established characteristics of TPVs, these long track vortices are expected to form via dynamic processes and occur in especially low shear environments (e.g., in the summer and in the high Arctic away from mid-latitude jets). Although these long-track TPVs are expected to possess behaviors somewhat different from an average TPV, they provide a more manageable case set through which to study the processes of vortex formation and movement. Further, an understanding of the physical and dynamical conditions that allow these vortices to persist for such extended periods may provide additional insight into

the behavior and predictability of TPVs as a whole. This, in turn, may improve the predictability of Arctic cyclones and other impactful weather events associated with TPVs. Long-lived anomalies are especially likely to impact surface weather, either by interacting with multiple Arctic cyclones or by entering the mid-latitudes.

## 2   Data and methods

### 2.1   TPVTrack 1.0

To study long-lived TPVs, we utilize TPV tracks generated with the TPVTrack 1.0 software, described in depth by Szapiro and Cavallo (2018). A brief description of TPVTrack highlighting the relevant features of the package is provided here for convenience. Compared with existing feature tracking systems, TPVTrack is intended to provide more detailed representations

of the spatial characteristics and temporal development of TPVs. TPVTrack utilizes a watershed segmentation technique to identify local extrema on the tropopause and group nearby grid points with the nearest associated minimum or maximum. This allows for the creation of TPV objects at each time step that can have irregular shapes, closely approximating true tropopause features. These objects are then tracked over time using horizontal and vertical correspondences between time steps to create unified TPV tracks, including a consideration of vortex splits and mergers. TPVTrack's efficacy as a tracking software and advantages over existing methods are verified by Szapiro and Cavallo (2018). Still, TPVTrack has noted limitations, including sensitivity to user-defined parameters and an elevated false-negative rate in temporal associations with respect to some existing methods. On the whole, though, TPVTrack provides an accurate case set of TPV tracks and characteristics ideal for this study.

With the TPVTrack settings used in this study, vortices are only tracked polewards of 30°N, as the tropopause does not correspond as well to the 2 PVU dynamic tropopause in the tropics. Since very few TPVs progress this far equatorward, though, this is not expected to affect the results of this study. In addition, the filter distance parameter (which dictates when extrema and their surrounding watersheds are considered distinct from other objects) is set to 300 km, the software's default value for tracking TPV-sized features. TPVTrack outputs an assortment of information on the tracked TPVs, including the location, minimum potential temperature, and circulation of the vortex over time. TPVTrack also provides the maximum amplitude of the TPV at each time step (henceforth simply referred to as amplitude), defined as the maximum potential temperature in the object minus the minimum potential temperature within the object. Although TPVTrack saves information on the exact shapes of the TPV objects, for simplicity, in this study we define the size of TPVs in terms of their equivalent radius (henceforth simply radius). This equivalent radius is the radius of a disk with equivalent area to the TPV object.

## 2.2 Long-track TPV selection and statistical methods

The most recent available set of long-term TPV tracks generated via TPVTrack was created using ERA-Interim data from 1979–2018 (Dee et al., 2011); for consistency, we utilize atmospheric data from the same reanalysis for other components of this study. Any anomalies of atmospheric data presented within this study are calculated against a 30-year (1981-2010) monthly climatology also generated using ERA-I. Within the TPVTrack-generated dataset of TPV tracks (i.e., cyclonic anomalies that formed north of 60°N), we define long-track TPVs to be those with a lifetime greater than or equal to the 95th percentile of all lifetimes to ensure that only the most persistent TPVs are included. This threshold corresponds to a total vortex lifetime of around two weeks as the minimum requirement for inclusion and results in a total long track TPV case set of 2157 tracks. To quantitatively assess radiative effects on TPV genesis, we utilize a standalone, longwave radiation version of the Rapid Radiative Transfer Model (RRTM-LW; Mlawer et al., 1997), a well-documented software. RRTM-LW is accessed through the CLIMLAB Python package (Rose, 2018).

Throughout the text, probability density maps are used to assess the spatial distribution of long-track TPVs. To evaluate statistical significance on these maps, we leverage Monte Carlo simulation techniques. For cases in which a composite of long-track TPV cases is compared against the full TPV record, we use a Monte Carlo climatology method. Ten thousand trials are conducted in which sets of random TPVs are drawn and subjected to the same analysis techniques as the long track set. The long track composite is then compared with a normalized mean of these trials, and the distribution of outcomes in the random

trials is used to establish significance. For cases in which a subset of long-track TPVs is compared with another subset or in which an atmospheric variable is compared to its climatology, we use a Monte Carlo permutation test approximation. In this case, ten thousand random trials are conducted in which all data points are randomly sorted into one of two categories (e.g., data and climatology) and then subjected to the same averaging as before. The correctly sorted composites are then compared to the results of these randomly sorted trials to assess significance. In the case of non-spatial probability distributions, we utilize the Kolmogorov-Smirnov (KS) test to establish significant differences (Massey Jr, 1951). These non-parametric tests allow us to more accurately assess significant patterns in spatial vortex distributions.

## 3 Results

### 3.1 Characteristics and spatial distributions of long-track TPVs

As mentioned in Sect. 2, the threshold used in this study to distinguish long-track TPVs is a lifetime of approximately two weeks. As a preliminary check of the qualities of these TPVs, we will compare probability distribution functions of long-track TPV properties to distributions for all TPVs. All corresponding pairs of probability distributions are found to be statistically distinct from each other using a KS-test (Massey Jr, 1951). On average, long-track TPVs last for around three weeks, with a few vortices persisting for up to 90 days (Fig. 1a). In contrast, the average TPV in the track file remains coherent for around five days. Across the full TPV record, no clear seasonal pattern emerges (Fig. 1b). On the other hand, long-track TPVs show a clear preference towards the summer, with a peak in occurrence in June and July. Physically, this trend is expected as summers will tend to have lower wind shear across the Arctic and fewer features propagating from the mid-latitudes that may disrupt radiative TPV strengthening. As expected due to a longer period of potential development, long-lived TPVs also appear to be larger and stronger than average. TPV amplitudes for long-track TPVs average around 10 K, compared to 4 K for the average vortices (Fig. 1c). This is especially notable given the large number of summer long-track TPVs, as the relatively high shortwave radiation during these months could tend to dampen TPV amplitudes (Cavallo and Hakim, 2013). Average radii fall around 300 km and 400 km for all TPVs and long-track TPVs, respectively, while the respective average circulations are around 25 $m^2$ $s^{-1}$ and 50 $m^2$ $s^{-1}$ (Fig. 1d,e). These anomalously high circulation values are indicative of a strong attendant flow, which in turn may correspond with increased impacts on weather events like Arctic cyclones compared with smaller, shorter lived vortices. Long-lived TPVs tend to remain slightly further poleward on average, occurring most frequently at 75°N as opposed to 65°N for the full TPV track file (Fig. 1f). Interestingly, though, the tails of each distribution extending into the mid-latitudes match up almost exactly, indicating that while long-track TPVs may spend a greater amount of their lifetime in the high Arctic, they are about as likely as any TPV to exit the Arctic eventually.

Turning to spatial distributions of long-track TPVs, we further investigate the characteristics implied in Fig. 1f. Long-lived TPVs are anomalously common over the Arctic ocean and Canadian archipelago and slightly less common than average over northern Europe and far northern North America (Fig. 2a). Long-track TPVs also appear more likely than average to form in the high Arctic, with fewer vortex genesis events than average in the 60°N–70°N range (Fig. 2b). The notable similarity between the whole-track spatial probability anomalies (Fig. 2a) and the genesis probabilities (Fig. 2b) suggests that these TPVs remain

roughly stagnant in the high Arctic for a large portion of their lifetimes, surviving in the low vertical shear environment. Nevertheless, examining the locations at which long-track TPVs reach their maximum amplitudes, common pathways out of the Arctic emerge (Fig. 2c). In particular, long-track TPVs seem to preferentially follow pathways southward across the Canadian archipelago and Hudson Bay and westward from eastern Siberia across the Northern Pacific. So, it is possible that long-lived TPVs achieve this designation not only by remaining in the high Arctic but also by resisting high shear in the mid-latitudes for exceptionally long time periods. Still, shearing appears likely to be a predominant lysis mechanism of long-track TPVs, along with diabatic PV destruction from latent heating, as the vortices are anomalously likely to dissipate over the North Pacific and North Atlantic Oceans, along known storm tracks (e.g., Wernli and Sprenger, 2007) and jet maxima (e.g., Christenson et al., 2017) (Fig. 2d). Similar spatial maxima of lysis frequency along storm tracks have been noted in previous studies (Muñoz et al., 2020; Portmann et al., 2021).

## 3.2 Long-track TPV genesis

Beyond understanding where long-track TPVs develop, we are interested in assessing exactly how these vortices form. This information could be especially beneficial from a long-term forecasting perspective and could prove applicable to all TPVs, regardless of lifetime. Many TPVs form via splits, in which an existing TPV deforms to a point where two distinct vortices with their own closed circulations materialize (Szapiro and Cavallo, 2018). So, it is likely that a large percentage of long-track TPVs also form in this manner. Note that when a vortex splits, TPVTrack associates the existing track with the one of the resulting vortices and begins a new track for the other. Although TPV splits are important, they are slightly easier to forecast using an adiabatic, advection-driven perspective of PV anomaly development. TPVs that do not form from existing TPVs are less understood; these may include existing PV anomalies that are transported into the Arctic or PV anomalies that generate directly via diabatic and frictional effects. To better understand these TPVs which do not form from splits, we separate long-track TPV cases into two categories: likely splits and non-likely splits. Determining definitively whether one vortex split has split from another and the exact time at which the split occurred is a difficult computational task. To that end, we utilize a distance threshold to sort TPVs into likely splits and non-likely splits. At the genesis of each long-track TPV, the distance from the genesis point to the nearest TPV at the previous time step is noted. The 80th percentile of all such distances is taken as an approximate likely split/non-likely split cutoff. This corresponds to a threshold distance of around 1200 km, which, for example, would be the distance between pre- and post-split TPVs of around 400 km in radius (Fig. 1d) moving at around 20 m s$^{-1}$. It is of course possible that some TPVs form via very large-scale splits, or emerge near an existing vortex without splitting from it, but for the purposes of the following analysis, this threshold is reasonable.

Long-track TPVs that generate via likely splits are most common over the high Arctic (Fig. 3a), closely mimicking the overall genesis probability density spatial pattern (Fig. 2b). This is consistent with the expectation that most TPVs form via splitting. Long-track TPVs formed via non-likely splits show a notably different spatial distribution, however, with several maxima scattered around the Arctic region (Fig. 3b). The difference between these two spatial distributions highlights several regions where long-track TPVs are more likely to fall into the non-likely split category (Fig. 3c). In particular, we will focus on three such regions that exhibited statistically significant differences: eastern Siberia, Alaska, and Greenland. Each of these

clusters is located in the vicinity of a climatological maximum in RWB activity (Peters and Waugh, 1996; Wernli and Sprenger, 2007). Given the established connection between RWB and PV anomaly creation, Rossby wave activity appears to be a likely candidate for TPV generation. Moreover, these regions each lie along or just downstream of major terrain features (i.e., the central Siberian plateau and Verkhoyansk Range, the Brooks Range, and the Greenland ice sheet, respectively). These surface features have the potential to affect TPV formation either via direct vorticity generation from down slope flow and pressure drag or via reinforcement of Rossby wave activity via terrain-induced standing waves (Jung and Rhines, 2007). To understand these cases more clearly, we isolate non-likely split genesis long-track TPVs for each region using latitude-longitude boxes corresponding to the areas of statistical significance, which can be found in Fig. 3c.

Due to the complexity of the synoptic conditions surrounding the genesis of many of these long-track TPVs, attempts to composite cases or identify patterns computationally proved inconclusive. Instead, individual cases are manually analyzed, described, and sorted into overarching synoptic pattern categories. This method, of course, relies on some subjective categorization; to that end, we have kept the possible categories broad. This way, an overview of common genesis mechanisms can be provided, paving the way for future, more automated studies. Based on common synoptic patterns across all three clusters, non-likely split long-track TPVs are visually sorted into four categories of formation (which will be discussed in more detail below). An idealized sketch of each of these formation mechanisms, including the areas in which TPVs were found to form, may be found in Fig. 4. The categories include cyclonic wave breaks (Fig. 4a), anticyclonic wave breaks (Fig. 4b), ridge building (Fig. 4c), and vortex splitting (Fig. 4d; i.e., highly stretched TPV splits). As mentioned, these categories were chosen to be adequately descriptive of the diversity of cases involved, while limiting the subjectivity of the sorting process. For this reason, for instance, Rossby wave breaks are classified based on shear but not on poleward versus equatorward breaking. Additionally, some cases are sorted into multiple categories (e.g., cyclonic wave breaking and ridge building) if both processes potentially affect the formation of the TPV. Further, note that the sketches provided in Fig. 4 are highly simplified depictions for the purpose of broadly characterizing TPV genesis, and real-world cases may involve complex synoptic patterns on a variety of scales.

Pre-genesis time-lagged maps (including time lags of -24, -12, and -0 hours) of tropopause conditions for all of the manually sorted cases from each cluster can be found in Appendix A. Developing large-scale Rossby waves are evident in many cases across all three clusters, while other cases exhibit relatively small scale features in the vicinity of vortex genesis (Figs. A1–A9). The Rossby waves present manifest in a variety of orientations and stages of development. Some retain a relatively straight north-south alignment of ridge and trough axes, while others are tilted such that these axes are nearly east-west oriented. Regardless, the key finding is that none of the TPVs appear to have generated in situ; all were formed via the rapid deformation, transport, and strengthening of existing tropopause PV anomalies, many of which originated in the mid-latitudes (Figs. A1–A9). The primary dynamic mechanisms that cause this transport and deformation are those given in Fig. 4. For conciseness, only selected example cases of each mechanism from each cluster are provided and discussed in the main text.

TPV formation via cyclonic or anticyclonic wave-breaking is straightforward, closely following previous literature on such RWB events (e.g., Thorncroft et al., 1993; Gabriel and Peters, 2008; Martius et al., 2010; Röthlisberger et al., 2018). The nascent TPVs may emerge as PV anomalies anywhere within the breaking trough. In many cases, a cyclonic PV anomaly which

exists within the trough prior to the RWB is split into multiple vortices during the course of the wave break; any of these newly formed anomalies that cross the 60°N threshold have the potential to become TPVs. For instance, in the example case from the eastern Siberia cluster (Fig. 5a-c), the most poleward closed contour becomes the feature of interest. In the Greenland example case (Fig. 5d-f), a vortex formed in the middle of the breaking filament produces the long-track TPV while in the Alaska example (Fig. 5g-i), a vortex on the southern end of the break remains poleward enough to classify as a TPV. The diversity of synoptic patterns present within these TPV genesis cases is also evident in the example cases. The wave breaks occur on different length scales, with different degrees of tilt, and at different latitudes, yet all eventually produce a coherent vortex north of 60°N (Fig. 5). The anicyclonic wave break examples present a similar trend; each case broadly matches the sheared wave break structure presented in Gabriel and Peters (2008), but with a variety of orientations and surrounding environments (Fig. 6).

The ridge building category is more novel and includes cases influenced by processes on a variety of scales. In these cases, the growth of a Rossby wave ridge (including shortwaves and longwaves) influences the formation of the TPV, although there is not clear evidence of a wave break in the traditional sense or orientation. In the early stages of these cases, a broad region of relatively low potential temperature on the dynamic tropopause exists to the north of a ridge (i.e., a region of high potential temperature). This broad region of low potential temperature may include several pre-existing embedded PV anomalies. As the pattern develops, the ridge builds poleward and outwards compressing the region of low potential temperature and pushing it northward. This continues until the broad high-PV region has fractured into distinct PV anomalies, some of which have been pushed into the Arctic by the growing ridge. The cause of the ridge growth likely varies by case. One potential catalyst is diabatic effects associated with surface cyclone development, while other cases may simply result from a shortwave ridge propagating through the region. The example cases for this mechanism illustrate the some of the many scales on which this process can occur (Fig. 7). The eastern Siberia example in particular shows the potential for several PV anomalies to result from one building ridge (Fig. 7a-c). The Alaska example, on the other hand, demonstrates the close connection between the ridge building and RWB mechanisms (Fig. 7g-i). In this case, while a CWB appears to be in progress, a large ridge builds poleward, further compressing the PV filament. The stretched TPV split cases are perhaps the most straightforward as they are really just likely-split events that exceeded the distance threshold. Because of the relative simplicity of these cases, explicit example cases are not included. In these, the new vortex emerges from a highly-deformed split from an existing polar-originating PV anomaly, with no clear evidence of Rossby wave or extra-Arctic influence. Because TPVTrack assigns the existing track to only one of the two resulting vortices, these splitting events are flagged as new TPV tracks.

Results of the non-likely split genesis categorization process for all three clusters are provided in Table 1. Across all three clusters, cyclonic wave breaking is the primary mode of TPV genesis, accounting for slightly under half of all genesis events (Table 1). Given that TPV genesis must occur at or north of 60°N, it is reasonable to expect that cyclonically sheared wave breaks are preferred, since, in many cases, the background zonal flow will be maximized south of 60°N. Interesting, then, are the notable numbers of anticyclonic RWB cases. Although this is the least common genesis mechanism in Greenland and Alaska, it is tied for second most common in eastern Siberia (Table 1). These cases require a rather poleward maximum in the mean zonal flow in order to facilitate TPV formation, a pattern which is perhaps more climatologically favorable over Asia.

These AWB events also appear slightly more likely to occur during the summer when the jet would be climatologically more poleward, though limited sample sizes hinder definitive statements on this point. Ridge building is the second most common genesis mechanism across all three clusters, making up around 30% of the total cases (Table 1). It is important to note here once again, though, that the ridge building category includes cases on a variety of scales. For instance, several eastern Siberia cases involve a shortwave ridge fracturing a small region of high PV, while many of the Alaska cases involve large-scale blocks. Stretched TPV split cases occur with some regularity in all three clusters, as was expected from our approximate likely split/non-likely split differentiation method. These tend to occur when an anomalously equatorward TPV begins to move northward back into the Arctic but is sheared apart in the process.

Links to surface cyclone activity in each of these clusters also cannot be ignored. The eastern Siberia and Alaska clusters could be influenced by cyclones in the north Pacific, while TPV formation in the Greenland region may be affected by north Atlantic storms. Surface cyclones have the ability to impact the characteristics of a RWB event and the resulting PV anomalies. Moreover, as suggested above, diabatic heating from surface cyclones may play a key role in the ridge building genesis mode, which appears especially likely in many of the Alaska cluster cases. Once again, it is important to note that these results are based on a visual inspection of the synoptic patterns provided in Figs. A1–A9 and are intended only to convey broad trends in TPV genesis mechanisms. Individual cases are often highly complex and may diverge slightly from the idealized patterns provided in Fig. 4.

To this point, the focus has been on kinematic influences on the genesis of non-likely split TPV cases. For completeness, we now briefly pivot to an analysis of the moisture anomalies surrounding the new TPVs, testing whether vertical moisture gradients may play a role in generating TPVs in a similar manner to which they intensify existing TPVs. PV can be generated via spatial gradients in diabatic tendencies over time, including radiative cooling by water vapor (Cavallo and Hakim, 2013). For this reason, a dry anomaly along and just above the tropopause can strengthen a TPV via the generation of PV. Across all three clusters, a dry anomaly near the tropopause develops locally within about one day of the eventual TPV genesis (Fig. 8). This is representative of the intrusion of dry stratospheric air to lower levels that would otherwise consist of tropospheric air as the tropopause begins to lower. There is no evidence of a preexisting moisture anomaly originating from elsewhere (e.g., higher in the stratosphere) and moving to the eventual TPV genesis location (Fig. 8). Thus, while the dry anomalies likely begin to contribute to the intensification of TPVs, they are quite weak around the time of genesis and it is unlikely that they result in a substantial longwave cooling anomaly to generate a TPV at these short time scales. In other words, the dry anomalies appear to be coincident with the strong dynamical forcing surrounding Rossby wave evolution as illustrated in Figure 4.

To further quantify this assertion, we use a series of simple experiments with RRTM-LW in a single column set-up. For each cluster, average humidity, temperature, and vorticity vertical profiles are taken at the genesis site 24 hours before TPV genesis occurs (i.e., vertical profiles through the centers of panels a, e, and i in Figure 8). All other absorbers are set to constant values, so that only water vapor varies with height and only water vapor will affect the vertical gradient of longwave cooling. The RRTM-LW calculated profiles of longwave cooling are then used with the average vorticity profiles to derive PV creation rates. If physical processes were able to fully explain TPV genesis, we would expect these PV creation rates to nearly match the actual observed change in PV over the one day period; however, for each cluster, the calculated tendency is much lower

than the observed change (Fig. 9). In fact, calculated PV tendencies based on longwave radiative effects are on the order of around 0.1 PVU day$^{-1}$, while the observed changes are on the order of 1 PVU day$^{-1}$ (Fig. 9). Thus, as expected dynamic processes must dominate during the period of TPV formation.

### 3.3 Long-track TPV development and motion

We now turn our attention to how long-track TPVs strengthen, move, and decay over the course of their lifetimes and how these developments vary with season. Composite wind anomaly fields around long-track TPVs can provide insight into both the structure of the vortices and the surrounding environments. At the time of genesis, generally symmetric cyclonic flow fields surround both summer and winter long-track TPVs, on average (Fig. 10a,c). A slight meridional asymmetry in the zonal flow is evident in both seasons, with slightly stronger flow (relative to climatology) on the southern flank of the TPVs. This may indicate some jet influence even very early in the TPV life cycle. It is also interesting to note that the wind anomalies are higher in the summer, when climatological wind speeds are lower, demonstrating that these TPVs possess relatively similar circulation strengths at genesis regardless of season. Moving forward in time to the time of vortex lysis, notable flow asymmetries appear in both seasonal composites, with flow on the southern flank becoming especially dominant (Fig. 10b,d). In the summer, this asymmetry is somewhat lower in magnitude, and the zonal wind anomalies remain zonally aligned (i.e., directly north and south of each other). Overall, this is indicative of TPVs that have strengthened but remained relatively stable over time. In contrast, the maximum anomalous westerly flow speed on the southern side of the winter composite is around 8 m s$^{-1}$ higher than the maximum anomalous easterly flow on the northern side (Fig. 10b,d). Moreover, the wind anomalies are not zonally aligned, with the southern side winds maximizing to the east of the vortex center. In contrast to the summer, this indicates TPVs that (in addition to becoming much stronger than the summer TPVs) have become asymmetric and sheared by the background flow. This may indicate that wintertime long-track TPVs are especially likely to dissipate by moving equatorward and interacting with the polar jet. Summer TPVs, on the other hand, may dissipate via shearing with a weaker, more poleward summertime jet or as a result of other processes in the high Arctic.

To examine long-track TPV development between vortex formation and dissipation, we normalize by the percentage of vortex lifetime completed, so that long-lived vortices of various time scales can be compared. The centered composites in Fig. 10 indicated that winter TPVs take on a more asymmetric structure over time, with relatively stronger flow developing on the southern flank of the vortex. Indeed, over the course of wintertime long-track TPV lifetimes, the median southern flank wind speed more than doubles, from 10 m s$^{-1}$ to 20 m s$^{-1}$ (Fig. 11a). This contrasts sharply with summertime cases, where median speeds increase only modestly over time. Moreover, southern flank flow speeds during the winter have a much higher ceiling —nearing 50 m s$^{-1}$ in some cases—than in the summer when most cases maximize at around 20 m s$^{-1}$ (Fig. 11a). Taken together, these results strongly suggest that winter long-track TPVs are influenced substantially by the presence of a strong jet that plays a role in the evolution and dissipation of these vortices.

Correspondingly, winter long-track TPVs as a whole progress further equatorward than summer cases, especially towards the end of their lifetimes (Fig. 11b). Both summer and winter TPVs gradually move equatorward over time, dropping southward by about 10° in latitude over the first 80% of their lifetime. From this point, the winter TPVs continue to progress equatorward

while the summer TPVs remain at a fairly constant latitude. Winter TPVs also seem more likely to progress extremely far south, with some cases approaching 40°N by the end of their lifetime (Fig. 11b). This southward progression is closely linked with the increasing southern flank wind speeds noted above. Both suggest that wintertime long-track TPVs are especially prone to exit the Arctic and are capable of surviving along a jet for long periods of time. Turning to the intrinsic characteristics of the TPVs, we find that wintertime vortex amplitudes are notably higher than in the summer (Fig. 11c). This disparity could be related to shortwave radiation or latent heating driven PV destruction in the summer (e.g., Cavallo and Hakim, 2013). Or, in some cases with especially strong TPVs, it may be reflective of the tropopause reaching all the way into deep wintertime boundary layer inversions at the apex of the TPVs strength. This hypothesis is also supported by the finding that long-track TPVs most often reach their maximum amplitude over land (Fig. 2c). During the winter, an inversion would often be present over this high-latitude land, while in the summer, surface inversions would be less likely to occur. Vortex radii are comparable between the seasons, with most long-track TPVs falling between 300 km and 500 km throughout their lifetimes (Fig. 11d). It is interesting to note though that median TPV amplitude and radius do not exhibit the same trend as southern flank zonal wind speed over time, even as these these TPVs begin to move under the influence of greater shear from their surroundings (Fig. 11a,c,d). It appears from this that during their time in the high Arctic, these TPVs may strengthen and stabilize to a point where they are less impacted by shearing and diabatic heating effects.

Throughout their lifetimes, long-track TPVs also tend to be, on average, associated with a surface cyclone (Fig. 12). During both the summer and the winter, at the time of TPV genesis a cyclone signal is already present in composites (Fig. 12a,c). This is not unexpected since many of these long track TPVs formed by splitting from other TPVs, which may have already contributed to the development of a surface system. In both seasons, the surface cyclone is offset to the east of the TPV, providing suitable conditions for cyclone strengthening. At the end of their lifetimes, summer long-track TPVs are still associated with a surface signal, which is more intense than at genesis time and slightly less offset from the TPV center (Fig. 12b). This nearly vertically-stacked pattern seems to show that summer TPVs tend to dissipate while part of mature cyclone systems. On the other hand, for winter cases at the time of lysis, the cyclone signal is offset from the TPV center by several hundred kilometers more than at genesis (Fig. 12d). In light of the findings presented above about winter long-track TPV development, this seems to indicate that winter TPVs often dissipate while actively interacting with a baroclinic zone to spur mid-latitude cyclogenesis. This, in turn, provides further evidence that long-lived TPVs in the winter tend to deteriorate due to interactions with mid-latitude jets and storm tracks.

Having analyzed the characteristics of long-track TPVs over time, we now move to an analysis of their motion. While long-track TPVs are in the high Arctic, they tend to move slowly and be advected within the background flow at the time. Of particular interest, then, are TPVs that exit the Arctic. As discussed earlier, these TPVs have the potential to influence a variety of high impact weather events in the mid-latitudes. Building on the definition that a TPV forms north of 60°N, we consider an Arctic exit to be any TPV that crosses this parallel and remains south of it for at least a day. To remove double counting TPVs that oscillate around 60°N, we include only the first Arctic exit for each track in the composites. Long-track TPVs can exit the Arctic in any part of the world, though they tend to do so preferentially in two regions: central Siberia and Canada (Fig. 13a). The Canadian exit pathway in particular is very narrow, with many TPVs entering the mid-latitudes near the Hudson Bay.

TPV exits occur throughout the year, with a slight uptick in frequency during the winter, when increased jet and Rossby wave activity may provide more exit opportunities (Fig. 13b).

On average, long-track TPVs survive for around two weeks after exiting the Arctic (Fig. 13c). As discussed above, this seems to be a unique feature of these very long lived vortices: the ability to persist for extended periods of time in the mid-latitudes despite higher ambient shear. TPVs exiting through Siberia have slightly longer lifetimes after exit than those exiting through Canada, hinting that these two sets of vortices behave differently after exiting the Arctic. Indeed while both groups of TPVs spend at least a few days in the mid-latitudes on average, the Canadian TPVs that survive past this point rapidly rebound into

the Arctic (Fig. 13d). The Siberian exits, on the other hand, remain south of 60°N for over a week on average. These trends align well with the TPV pathways visible in Fig. 2c. The Canadian exits tend to move over the Hudson Bay and into the North Atlantic before either dissipating or curving northward near Greenland. The Siberian exits begin to move due eastward over Asia and the North Pacific, with some reaching the western coast of North America before dissipating or turning northward again. In general, vortices from both exit pathways gradually weaken upon entering the mid-latitudes, though those that reenter

the Arctic appear capable of strengthening once again (Fig. 13e,f).

To better understand the spatial characteristics of TPVs exiting the Arctic, we examine the average motion of all long-track TPVs across the Northern Hemisphere. Over the Arctic Ocean, average TPV movement is relatively slow and not in any dominant direction, as expected given how long many of these TPVs seem to linger in the high Arctic (Fig. 14a). The movement of TPVs in this regime is best understood using vortex and advection dynamics, as the predominant background

flow at a given time will tend to be the main driver of TPV propagation. In the mid-latitudes, on the other hand, TPVs move relatively quickly and are clearly controlled by climatological westerlies and larger-scale flow. At this point, the movement of TPVs will become more dominated by Rossby wave dynamics. In addition, the two primary Arctic exit regions discussed above (over Canada and Siberia) are salient in the average motion composites (Fig. 14a). As seen in Fig. 13a, the Canadian exit corridor is compact, while the Siberian pathway is far more diffuse. TPVs exiting over Canada do so quickly, reaching average

speeds of up to 7.5 m s$^{-1}$ while moving predominately north to south (Fig. 14a). Siberian exits, though, move more slowly equatorward, tracking gradually to the southeast over Russia towards the Pacific. The exact conditions which trigger an Arctic exit, therefore transitioning the vortex from an advection-driven regime into a wave-driven one, are not yet well understood and merit further study.

Depending on how far into the mid-latitudes they progress, these TPVs may either reenter the Arctic quickly or move zonally

while remaining at lower latitudes. Considering TPVs that exit the Arctic and remain in the mid-latitudes for at least two days, 57% of Canadian exits and 45% of Siberian exits eventually reenter the Arctic. Arctic reentry patterns for the two groups mirror the trends seen in Fig. 13d (Fig. 14a). Canadian exits which reenter the Arctic do so relatively quickly, turning sharply over the North Atlantic Ocean and moving poleward over Greenland. TPVs which progress southward of 50°N, however, appear more likely to remain in the mid-latitudes, moving across the North Atlantic and into Europe. In contrast, on average,

most Siberian-exiting TPVs move across the Pacific towards North America. Some of these vortices may reenter the Arctic over eastern Siberia or Alaska, though a notable number of streamlines progress all the way to the United States and Canada. These TPVs that remain outside of the Arctic and progress into Europe and North America may be especially important to

the development of downstream mid-latitude systems. A schematic view of these Arctic exit pathways provided in Fig. 14b highlights the different patterns of TPV motion between these two regions.

## 4    Conclusions

Using TPV tracks generated via the TPVTrack software from ERA-I data, this study isolated a set of the longest-lived TPVs from 1979 through 2018 (specifically those with a lifetime of longer than two weeks). These TPVs were found to be on average larger and stronger than other TPVs, likely owing to their extended window of development. These TPVs are also found to reside at higher latitudes for a large portion of their lifetimes and occur preferentially in the summer. These findings are consistent with the notion that TPVs persist and thrive in low shear environments. Wintertime long-track TPVs exhibit substantially higher amplitudes on average than summer cases. This could be related to shortwave radiation or latent heating forcing during the summer, though Cavallo and Hakim (2013) found the contribution of shortwave radiation to be small in general. On the other hand, in a few cases, this wintertime amplitude spike may be a product of the tropopause folding down to the boundary layer within the strongest of these TPVs. In this case, strong boundary layer inversions in the winter could account for the relatively high vortex amplitudes on the tropopause, while these inversions would not be as noticeable during the summer. We find that long-track TPVs tend to reach their maximum around 40-50% of the way through their lifetime, which is consistent with the findings of Kew et al. (2010). However, unlike the general anomalies assessed in that study, which begin to decline in amplitude immediately at this point, the long-track TPVs analyzed here remain roughly stable until around 80% of the way through their lifetime. This is especially in conjunction with the finding that many of these long-track TPVs seem to reach their maximum amplitude while exiting the Arctic (Fig. 2c). Thus, a substantial number of long-lived vortices seem to be able to persist for periods of longer than a week even after exiting the Arctic. Understanding the vortex structure that allows these vortices to persist even in the presence of high ambient shear certainly merits further study.

We also find a notable seasonal dichotomy between the average life cycle and characteristics of long-lived vortices. Summer long-track TPVs tend to thrive in the low shear environment of the summertime Arctic, remaining relatively more poleward than their wintertime counterparts, especially towards the end of their lifetimes. As a result, these vortices retain a more symmetrical structure throughout their life cycle. These summer long track vortices are notable, as they are especially likely to impact Arctic cyclone activity and contribute to sea ice break up. In fact, many of these long-lived vortices persist for long enough to influence several different Arctic cyclones throughout their lifetime. Lysis mechanisms for these summer TPVs are not immediately obvious. It is likely that some of the vortices dynamically dissipate either via absorption into a weak jet or deformation over the Arctic ocean. Others, perhaps, are weakened during their interactions with strong Arctic cyclones, as middle troposphere latent heating has the potential to destroy PV above. In contrast, winter long-track TPVs appear to be heavily influenced by the climatologically stronger polar jet during this season. These winter cases tend to become especially strong due to low incoming shortwave radiation in the high Arctic before progressing southward. As the TPVs move into the mid-latitudes, they gradually encounter greater shear on their southern flank (indicative of interaction with the polar jet) and

yet are able to persist in the mid-latitudes for a week or more in some cases. This persistence along the jet may make these long-lived TPVs especially consequential in terms of Rossby wave initiation and surface cyclogenesis potential.

Despite the evident preference for winter TPVs to move equatorward and interact with the jet, long-track TPVs from all seasons commonly exit the Arctic. These Arctic exits are found to occur through two main pathways: through Siberia and through Canada. Siberian exits tend to survive for slightly longer after exiting the Arctic, and move westward across Asia and the northern Pacific Ocean towards North America. Canadian exits, on the other hand, tend to quickly reenter the Arctic over Greenland or, in some cases, move across the North Atlantic into Europe. Both of these exit pathways are important from a forecasting perspective. TPVs moving down through Canada may be associated with cold air outbreaks over the North American continent, while TPVs that exit through either pathway and remain in the mid-latitudes may have downstream impacts on cyclones and other significant weather over North America and Europe. Also of note, the circular TPV exit and reentry pathways mirror the two climatological nodes of the tropospheric polar vortex (Waugh et al., 2017). Although this large-scale feature would only be present during the winter, it still serves as an elucidating example of how TPV movement is governed by the large scale flow.

In order to fully leverage the conservation properties of TPVs in terms of medium to extended range forecasting potential, it is also necessary to understand where and how these vortices tend to form. For the majority of long-track TPV cases, vortex genesis occurs in the high Arctic as a split from a previously existing TPV. These split cases are often kinematically driven, owing to shearing and deformation surrounding the TPV. Although not analyzed in depth here, these splitting TPVs at high latitudes are important for the maintenance of Arctic cyclones. A smaller subset of long-lived TPVs do not form from existing Arctic vortices. Rather, they appear to be linked to RWB events and building ridges, which fracture mid-latitude PV features and push them into the Arctic. These non-likely split cases occur preferentially in eastern Siberia, Alaska, and Greenland. These regions are in close proximity to noted climatological maxima in RWB events, although areas of maximum TPV genesis and areas of maximum RWB activity do not line up exactly (Peters and Waugh, 1996; Wernli and Sprenger, 2007). This may result from the fact that the PV anomalies formed by RWB also need to enter the Arctic in order to be considered TPVs. Perhaps, then, certain regions are conducive not only to wave breaking, but also to flow regimes that allow for easy advection of the resulting PV features into the Arctic.

Overall, the results of this study serve to confirm several existing physical characteristics of TPVs (such as their affinity for low shear environments) while also presenting new insights on their mechanisms of formation and behavior in different seasons. We hope that these findings will improve our ability to forecast TPV activity, and, by extension, cyclones and other high-impact weather events. Although these findings are specifically applicable to the small set of long-lived TPVs studied here, many of the principles of formation and movement are likely to apply to TPVs as a whole, albeit on the shorter time scales in which these regular vortices tend to exist. Furthermore, while this study has provided an overview of the genesis, development, and motion of long-lived TPVs, future work is needed to expand upon these findings. Numerical modeling studies could be used to quantify the relative contributions of dynamic and moisture forcings on TPV genesis. A more expansive study of TPV genesis mechanisms could elucidate new patterns of vortex formation and provide a more detailed look at the physical processes that precede vortex splitting. Such a study could focus on creating a more quantitative, objective set of TPV genesis mechanisms.

As an example, machine learning techniques or self-organizing maps may be a useful approach for objectively identifying genesis patterns, though that is outside the scope of this introductory examination. Future studies are also needed to translate these findings directly into forecasting applications. For instance, a more detailed understanding of how TPVs that form from RWB events go on to support Arctic cyclone development or why vortices sometimes exit the Arctic would greatly improve the forecasting potential of TPVs.

*Code and data availability.* ERA-Interim data may be freely obtained from https://apps.ecmwf.int/datasets/. The TPVTrack source code including a user manual is available at https://github.com/nickszap/tpvTrack.

*Author contributions.* SC and MB developed the project and methodology. MB carried out the investigation and visualization portions of the study under the supervision and guidance of SC. MB prepared the manuscript with contributions from SC.

*Competing interests.* The authors declare that they have no conflict of interest.

*Acknowledgements.* We would like to thank the University of Oklahoma Arctic and Antarctic Atmospheric Research Group for their contributions throughout this project. Additionally, we would like to thank Dr. Jason Furtado, who provided feedback on an early version of this project and its statistical methods. This work was supported under Office of Naval Research Grant #N00014-16-1-2489 and is based on work supported by the National Science Foundation Graduate Research Fellowship Program under Grant No. 105411900. Any opinions, findings, and conclusions or recommendations expressed in this material are those of the author(s) and do not necessarily reflect the views of
the National Science Foundation.

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

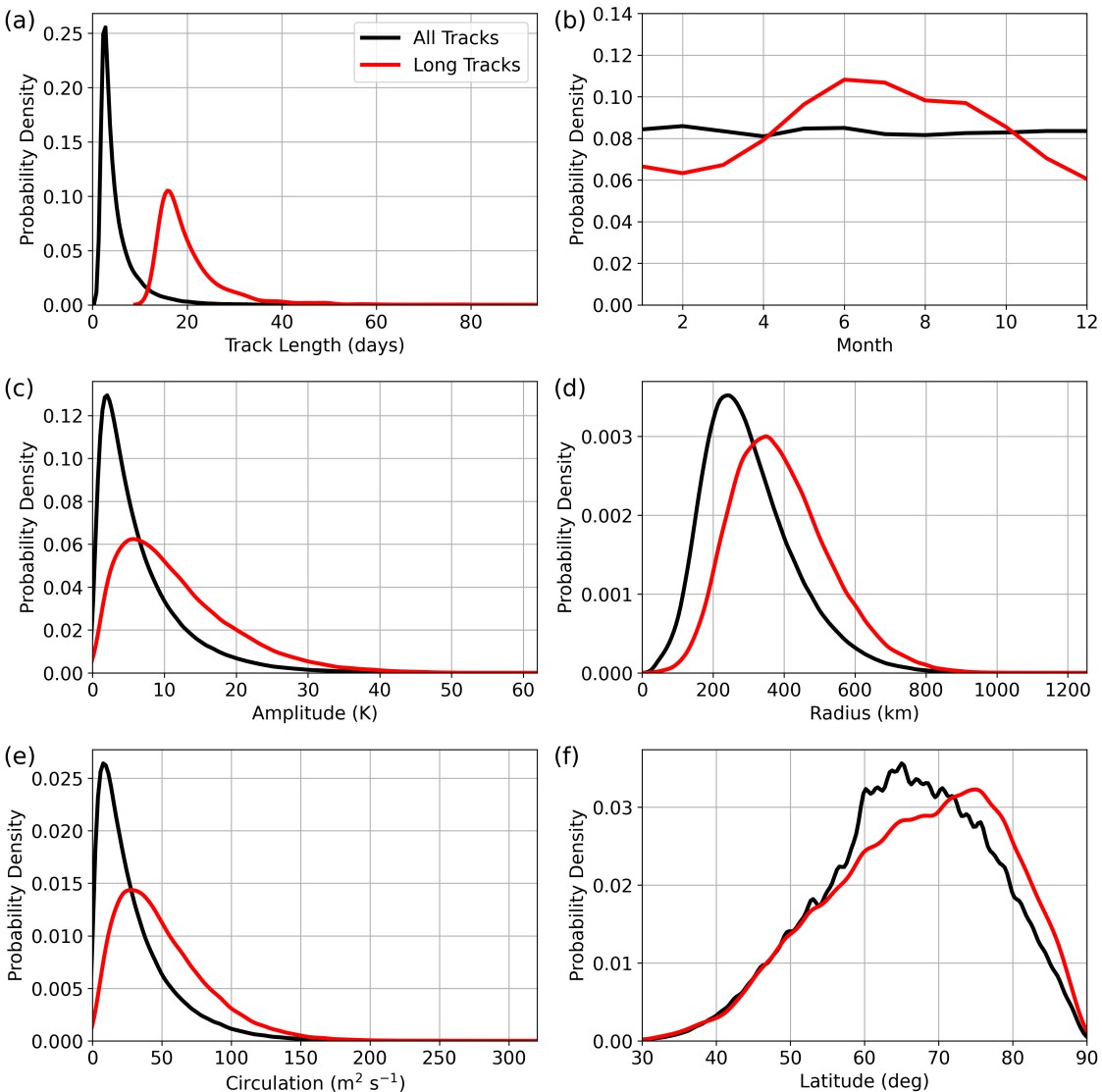

**Figure 1.** (a) Probability density distributions of TPV track length (days) for all TPVs (black) and all long-track TPVs (red). (b) As in (a), but for month of TPV occurrence. (c) As in (a), but for TPV amplitude (K). (d) As in (a), but for TPV radius (km). (e) As in (a), but for TPV circulation ($m^2$ $s^{-1}$). (f) As in (a), but for TPV latitudes (°N). For (b)-(f), all track points for each TPV are included individually. All pairs of distributions are found to be unique from each other at $p<0.05$ using a KS-test.

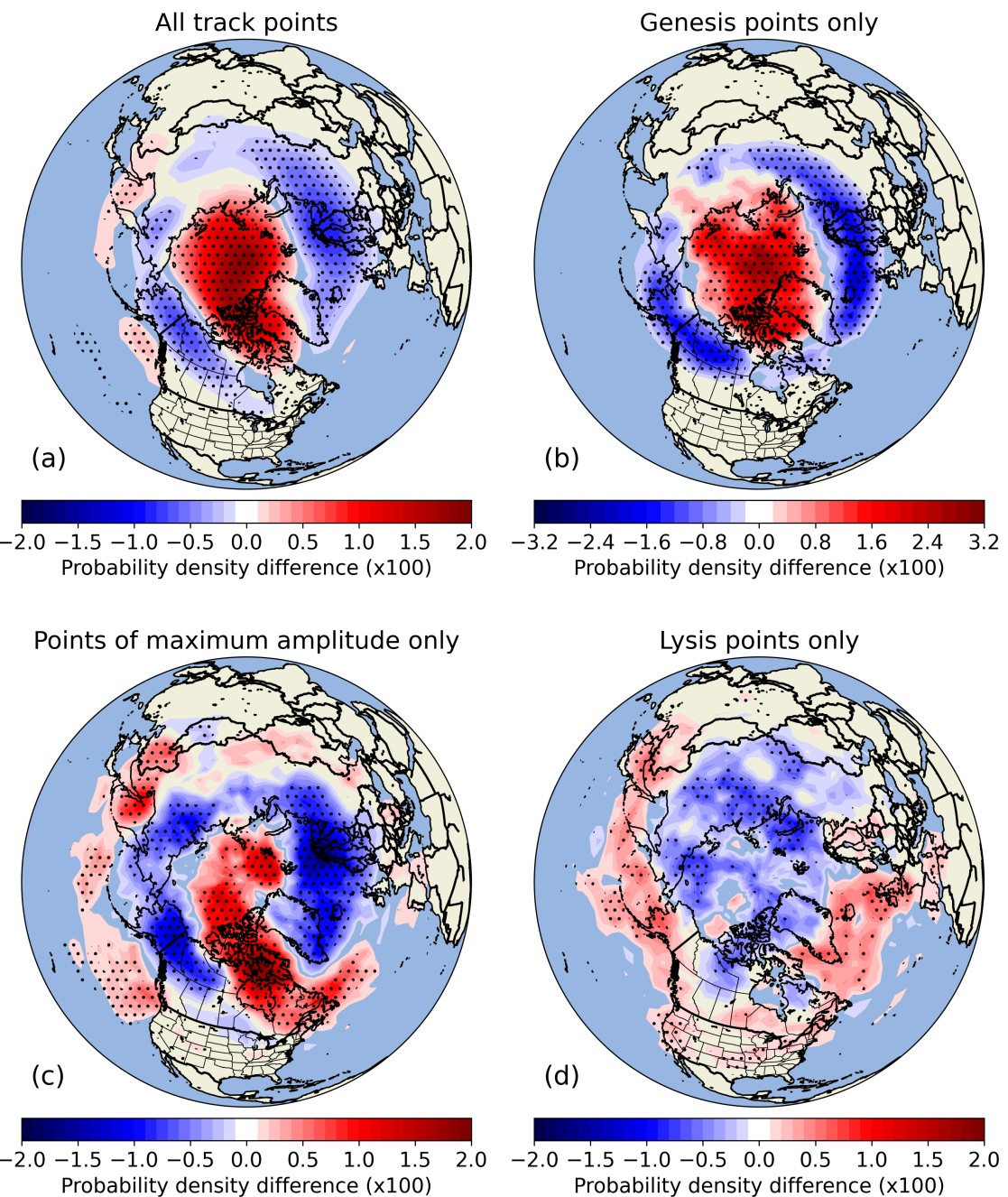

**Figure 2.** (a) Spatial probability density difference between all long-track TPV track points and all TPV track points regardless of track length (i.e., positive values indicate that a long-track TPV is relatively more likely to occur at that location). Stippling indicates significance at the 95% level established using the Monte Carlo climatology method described in the text. (b) As in (a), but showing the difference between long-track TPV genesis points and all TPV genesis points. (c) As in (a), but showing the difference between long-track TPV maximum amplitude points and all TPV maximum amplitude points. (d) As in (a), but showing the difference between long-track TPV lysis points and all TPV lysis points.

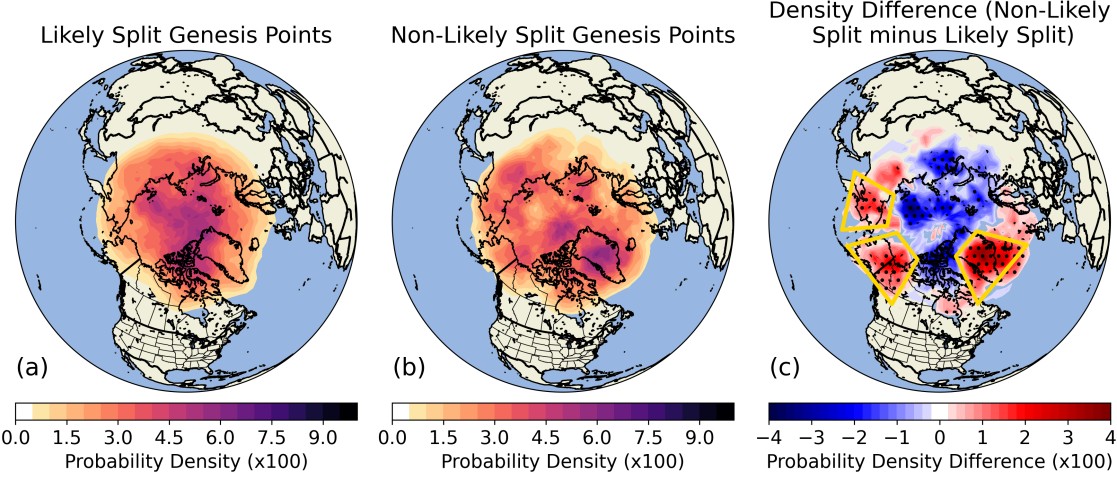

**Figure 3.** (a) Probability density distribution of all long-track TPV genesis points that formed via a likely split (as defined in the text). (b) As in (a) but for all long-track TPV genesis points that did not from via a likely split. (c) The difference between panel (b) and panel (a). Stippling indicates significance at the 95% level established using the Monte Carlo permutation test method described in the text.

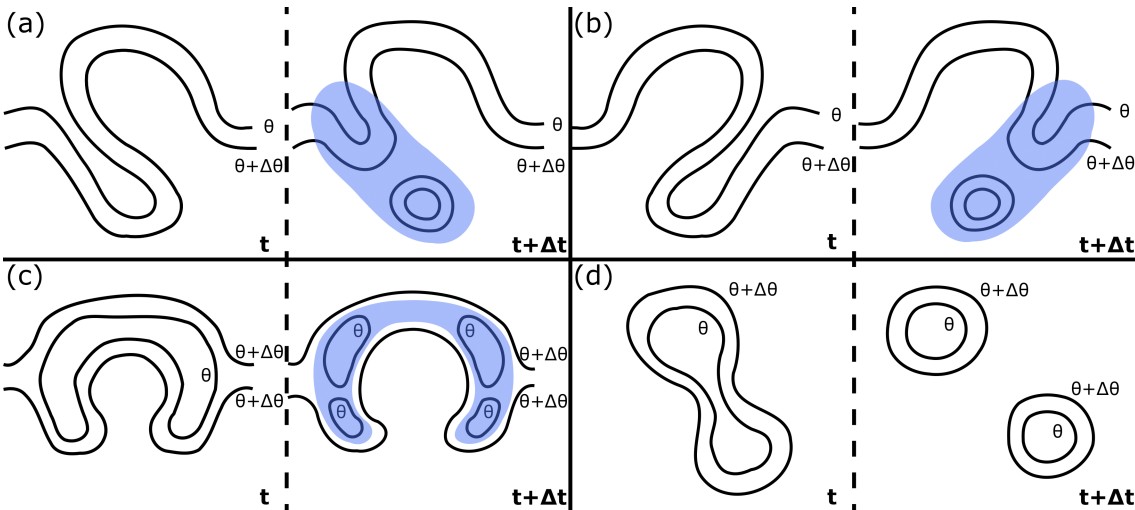

**Figure 4.** An idealized conceptual model illustrating the four major long-track TPV genesis mechanisms discussed in the text. Black contours represent hypothetical simplified isentropes on the dynamic tropopause, and consecutive panels show the evolution of these isentropes during TPV genesis. Blue shading represents the area in which the TPV of interest may be found in any given case. (a) Cyclonic wave breaking. (b) Anticyclonic wave breaking. (c) Ridge building. (d) Vortex splitting.

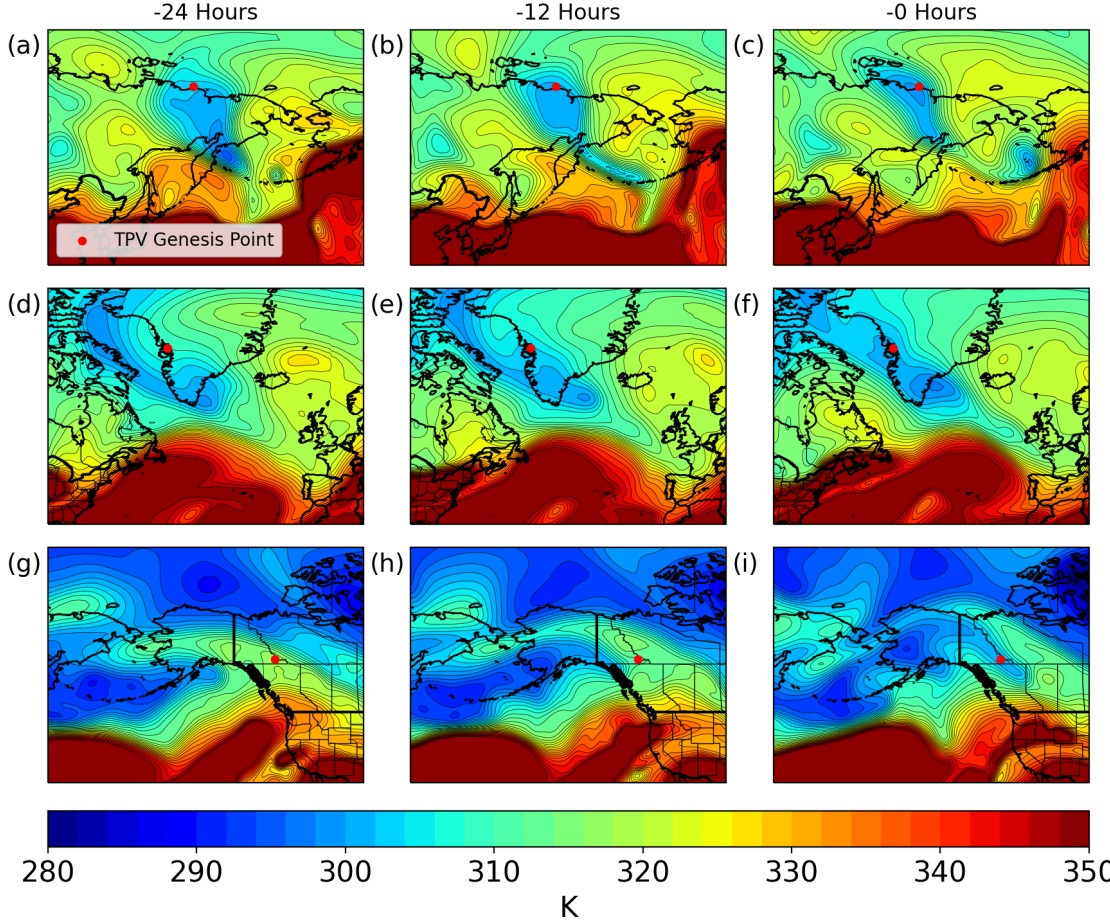

**Figure 5.** Example cases of the cyclonic wave breaking genesis mechanism from each of the three non-likely split clusters. Each panel includes a map of tropopause potential temperature (K; fill and contours) with a red dot to indicate the genesis location of the TPV. (a)-(c) Example case for the eastern Siberia cluster from September 11, 2005 with panels showing tropopause conditions 24 hours before, 12 hours before, and at the time of the genesis event. (d)-(f) As in (a)-(c) but for a September 10, 1993 case from the Greenland cluster. (g)-(i) As in (a)-(c) but for a November 8, 2007 case from the Alaska cluster.

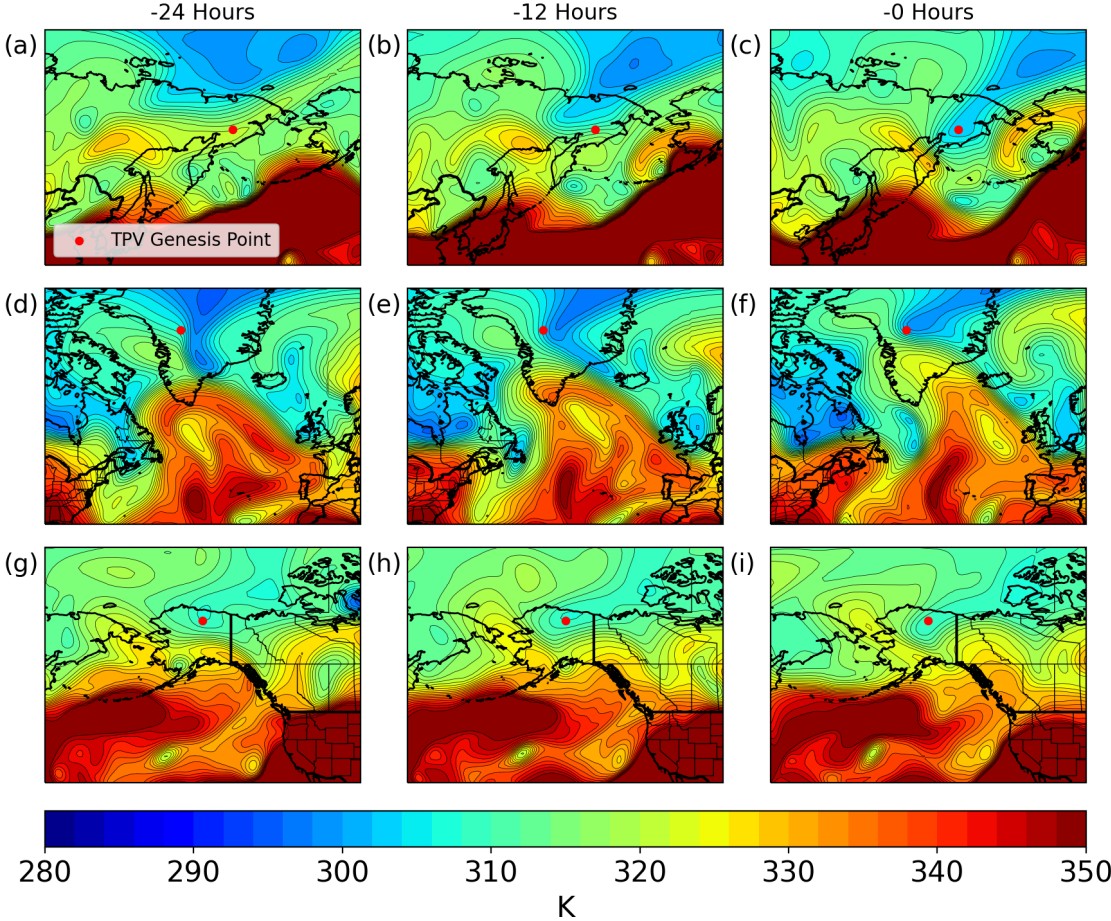

**Figure 6.** Example cases of the anticyclonic wave breaking genesis mechanism from each of the three non-likely split clusters. Each panel includes a map of tropopause potential temperature (K; fill and contours) with a red dot to indicate the genesis location of the TPV. (a)-(c) Example case for the eastern Siberia cluster from September 19, 1988 with panels showing tropopause conditions 24 hours before, 12 hours before, and at the time of the genesis event. (d)-(f) As in (a)-(c) but for a May 18, 2000 case from the Greenland cluster. (g)-(i) As in (a)-(c) but for a July 30, 1980 case from the Alaska cluster.

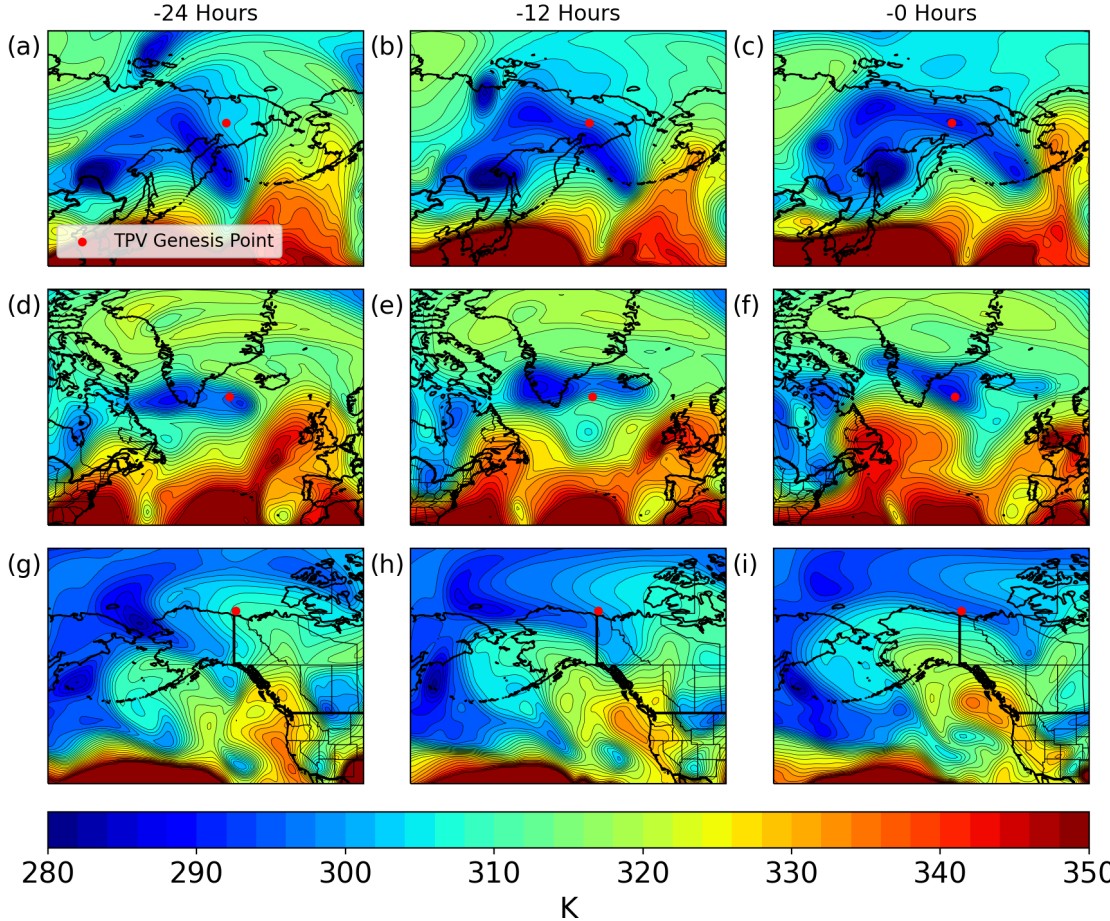

**Figure 7.** Example cases of the ridge building genesis mechanism from each of the three non-likely split clusters. Each panel includes a map of tropopause potential temperature (K; fill and contours) with a red dot to indicate the genesis location of the TPV. (a)-(c) Example case for the eastern Siberia cluster from November 10, 1996 with panels showing tropopause conditions 24 hours before, 12 hours before, and at the time of the genesis event. (d)-(f) As in (a)-(c) but for a September 16, 1998 case from the Greenland cluster. (g)-(i) As in (a)-(c) but for a November 21, 2000 case from the Alaska cluster.

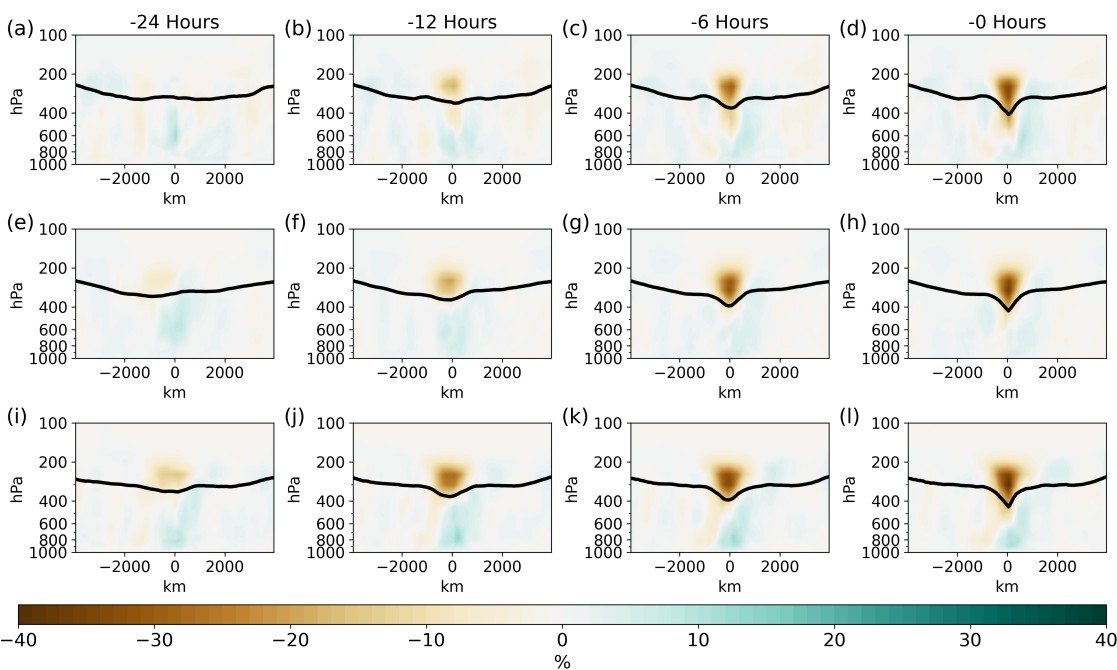

**Figure 8.** Rotationally averaged cross-section composites of relative humidity anomaly (%; fill) through the genesis points of all TPVs within the eastern Siberia non-likely split cluster at (a) 24 hours before TPV genesis, (b) 12 hours before TPV genesis, (c) 6 hours before TPV genesis, and (d) the time of TPV genesis. The 2 PVU surface (hPa) is included on each panel as a solid black line. (e)-(h) As in (a)-(d) but for the central Greenland non-likely split cluster. (i)-(l) As in (a)-(d) but for the northern Alaska non-likely split cluster.

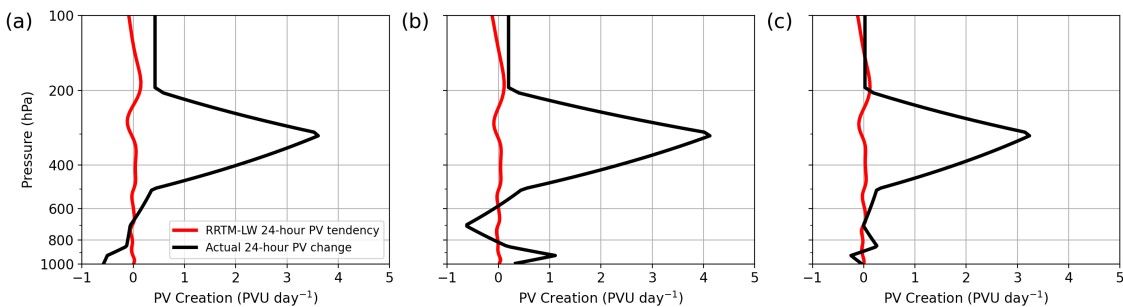

**Figure 9.** (a) Vertical profiles of true change in PV over the 24 hours preceding TPV genesis (black; PVU day$^{-1}$) and calculated change in PV during this time based on longwave heating and cooling from water vapor (red; PVU day$^{-1}$) averaged across all cases in the eastern Siberia non-likely split cluster. Expected changes in PV due to longwave heating and cooling are calculated using single column RRTM-LW with the average humidity and temperature profiles through the location of TPV genesis, 24 hours before the genesis event (i.e., a profile through the center of Figure 8a). (b) As in (a), but for the Greenland non-likely split cluster. (c) As in (a), but for the Alaska non-likely split cluster.

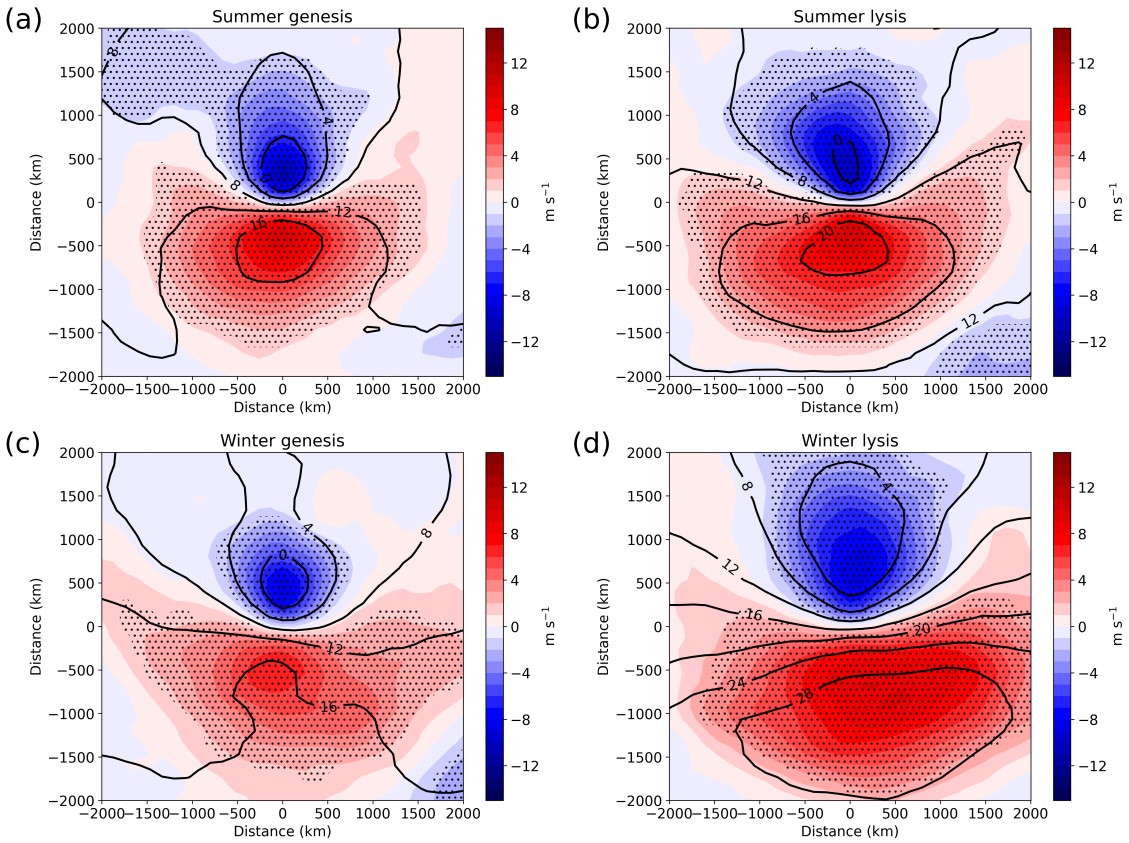

**Figure 10.** (a) TPV-centered composite of tropopause u-wind anomaly (m s$^{-1}$; fill) and u-wind (m s$^{-1}$; contour) at the time of vortex genesis for all long-track TPVs that form during the summer (JJA). Stippling indicates significance of the anomaly field at the 95% level established using the Monte Carlo permutation test method described in the text. (b) As in (a), but at the time of vortex lysis. (c) As in (a), but at the time of vortex genesis for all long-track TPVs that form during the winter (DJF). (d) As in (a), but at the time of vortex lysis for all long-track TPVs that form during the winter.

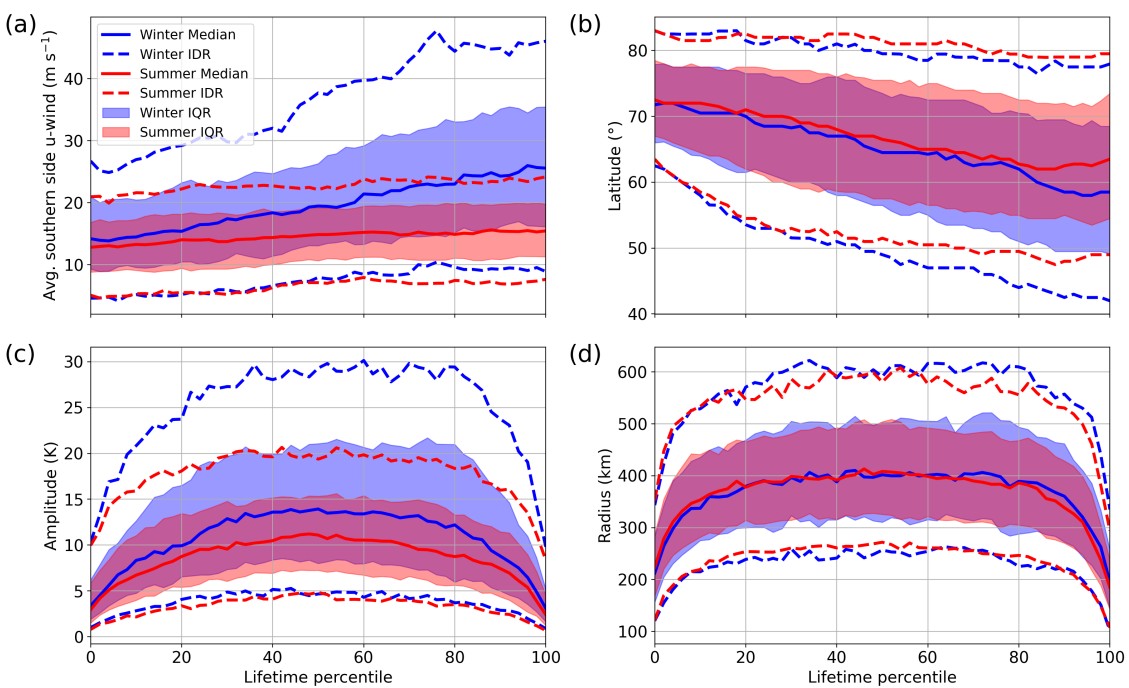

**Figure 11.** (a) Evolution of the average u-wind magnitude (m s$^{-1}$) on the southern side of the TPV center (defined as the bottom half of the TPV-centered boxes shown in Fig. 10) by the percentage of the total TPV lifetime that has been completed, averaged over all summer (red) and winter (blue) long-track TPV cases. Solid lines show the median value, shading represents the interquartile range, and dashed lines indicate the interdecile range. (b) As in (a) but for the latitude of the TPVs (°). (c) As in (a), but for the amplitude of the TPVs (K). (d) As in (a), but for the radius of the TPVs (km).

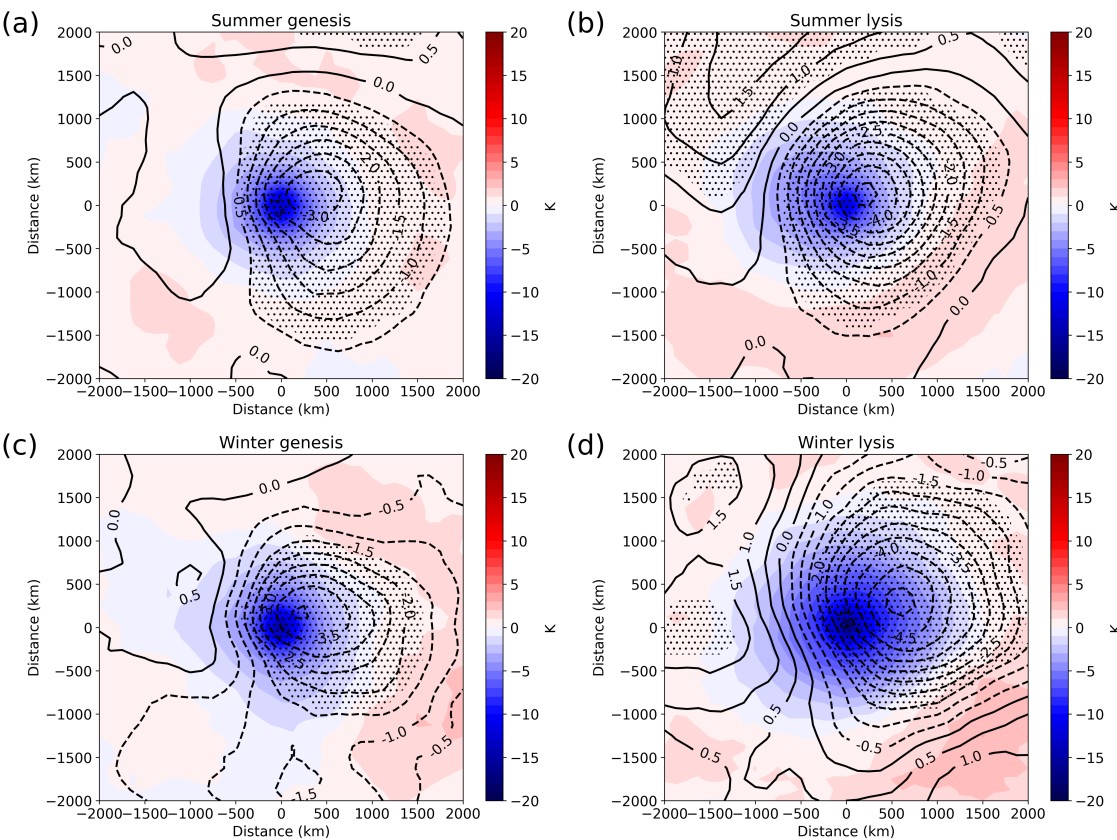

**Figure 12.** (a) TPV-centered composite of tropopause potential temperature anomaly (K; fill) and mean sea-level pressure anomaly (hPa; contour) at the time of vortex genesis for all long-track TPVs that form during the summer (JJA). Stippling indicates significance of the mean sea-level pressure pattern at the 95% level established using the Monte Carlo permutation test method described in the text. (b) As in (a), but at the time of vortex lysis. (c) As in (a), but at the time of vortex genesis for all long-track TPVs that form during the winter (DJF). (d) As in (a), but at the time of vortex lysis for all long-track TPVs that form during the winter.

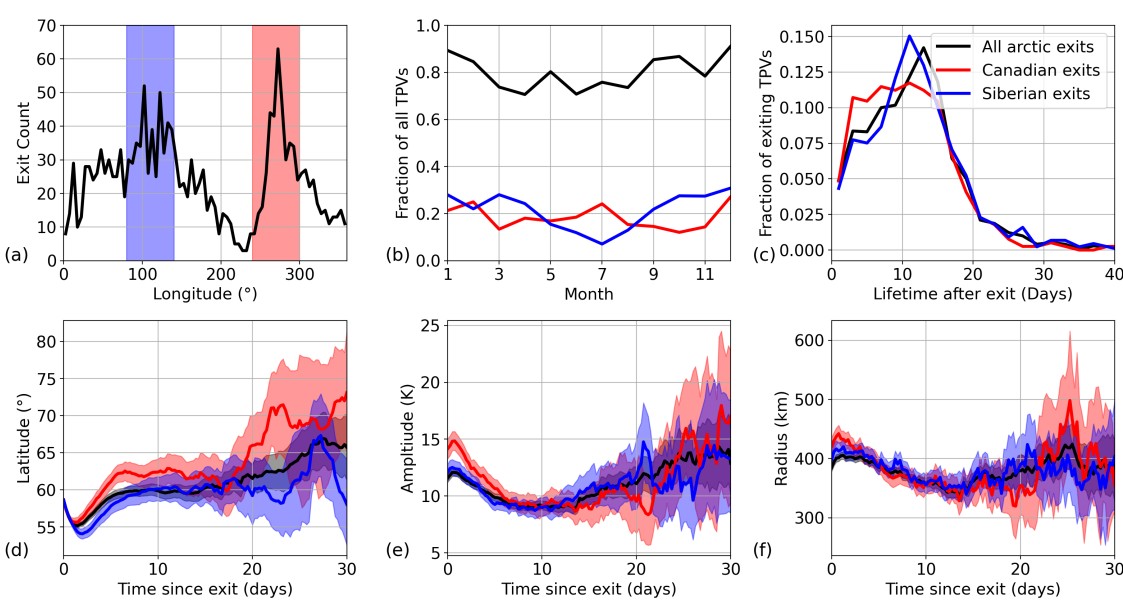

**Figure 13.** (a) Line histogram of longitudes (°) at which long-track TPVs exited the Arctic (crossed 60°N) for the first time. Blue shading represents the range of cases included in the Siberian exit maximum (80°– 140°), and red shading represents the Canadian exit maximum (240°– 300°). (b) The fraction of all long-track TPVs that exited the Arctic at any point by month of exit (black), along with the fraction of these TPVs that exited via the Siberian (blue) and Canadian (red) pathways. (c) Lifetime (days) of all exiting long-track TPVs (black), Siberian exits (blue), and Canadian exits (red) after exiting the Arctic. (d) Average latitude (°) of post-exit TPVs for all exits (black), Siberian exits (blue), and Canadian exits (red). Shading represents confidence at the 95% level established with bootstrap resampling tests. (e) As in (d), but for average TPV amplitude (K). (f) As in (d), but for average TPV radius (km).

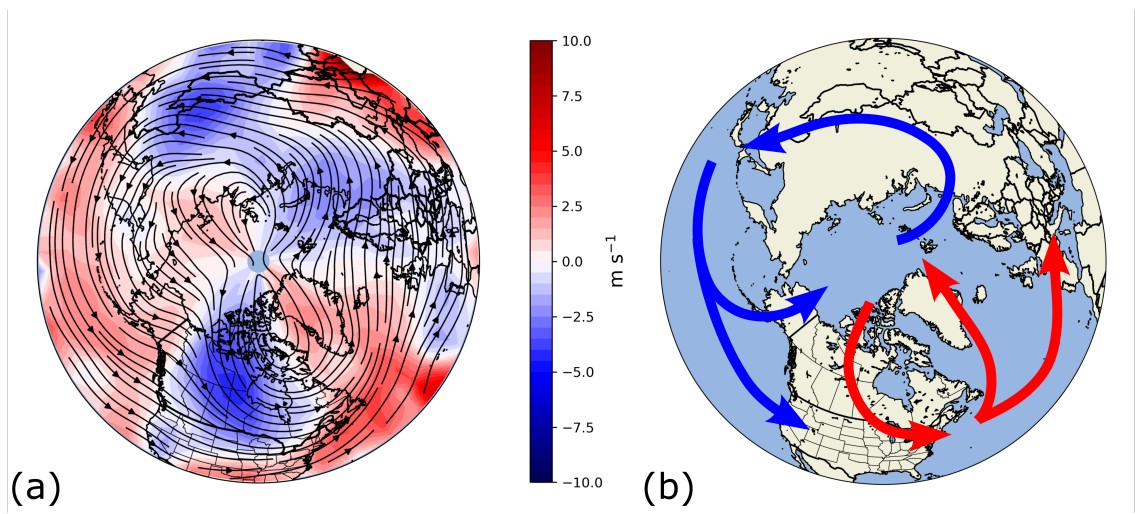

**Figure 14.** (a) Streamlines (black) of average long-track TPV motion and average meridional TPV speed (m s$^{-1}$; fill), calculated as vector averages within 5°by 5°latitude-longitude bins. (b) Schematic of the Siberian (blue) and Canadian (red) Arctic exit, transport, and reentry pathways observable in (a) and in Fig. 11

**Table 1.** Classification of non-likely split TPV genesis events for each of the three major clusters highlighted in Fig. 3 into the four categories discussed in the text and presented in Fig. 8. Note that some cases are sorted into two different categories (e.g., cyclonic wave breaking and ridge building). These classifications were made based on the tropopause potential maps provided for each cluster.

| Cluster (number of cases) | Cyclonic Wave Breaking | Anticyclonic Wave Breaking | Ridge Building | Stretched TPV Split |
|---|---|---|---|---|
| Eastern Siberia (24) | 10 | 7 | 7 | 4 |
| Greenland (63) | 30 | 8 | 17 | 14 |
| Alaska (31) | 14 | 4 | 9 | 6 |

## Appendix A: Complete Non-likely Split Genesis Figures

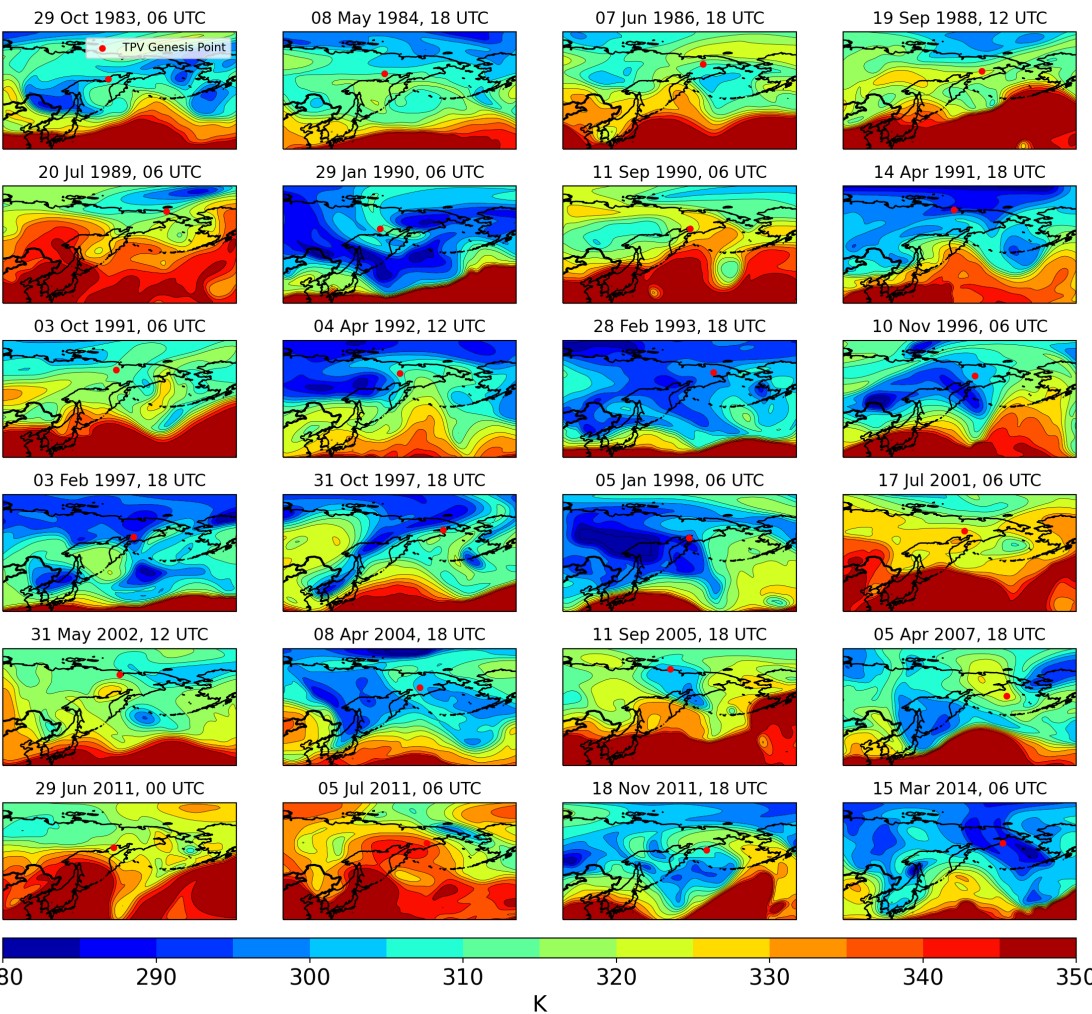

**Figure A1.** Maps of tropopause potential temperature (K; fill and contours) 24 hours prior to each TPV genesis event in the eastern Siberia non-likely split cluster. Red dots indicate the eventual genesis location of the TPV.

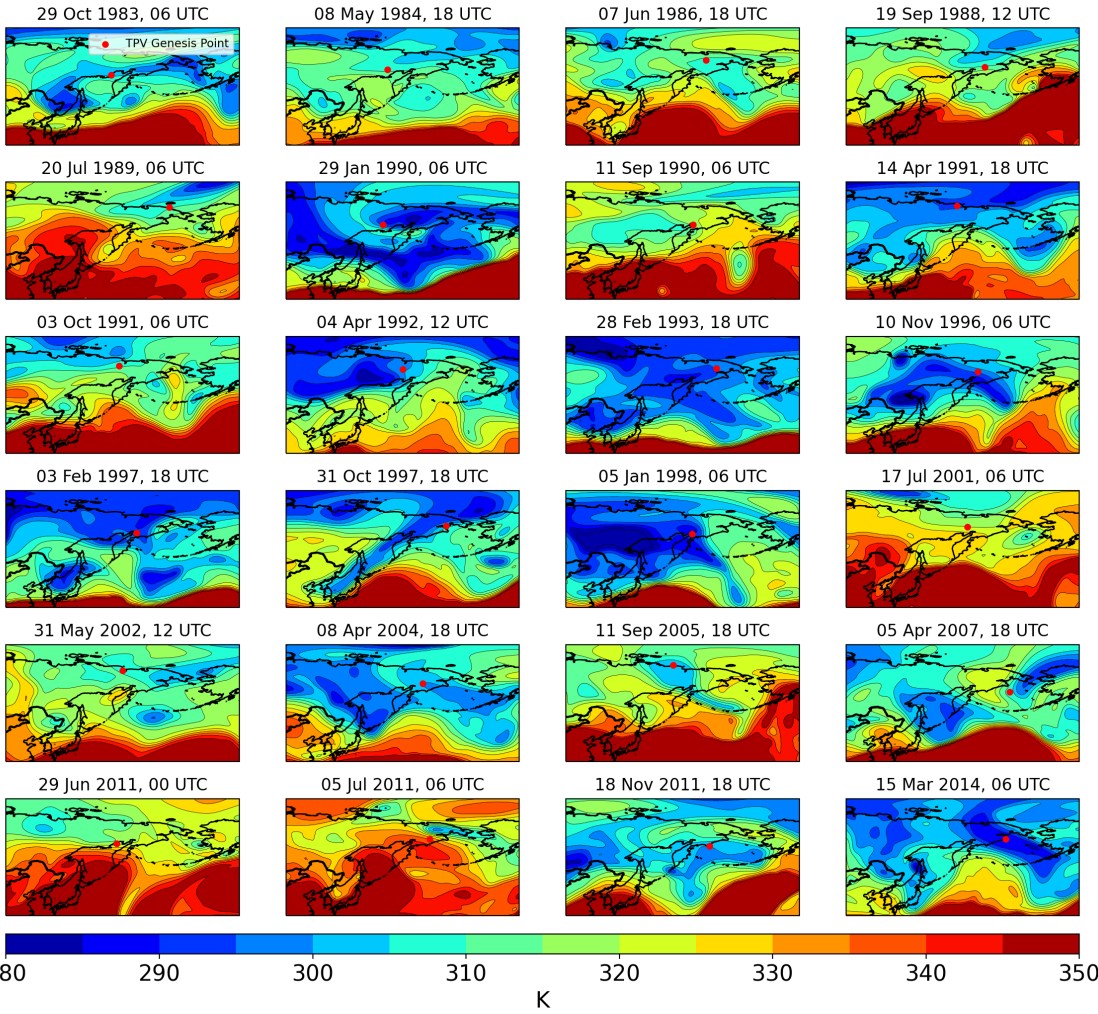

**Figure A2.** Maps of tropopause potential temperature (K; fill and contours) 12 hours prior to each TPV genesis event in the eastern Siberia non-likely split cluster. Red dots indicate the eventual genesis location of the TPV.

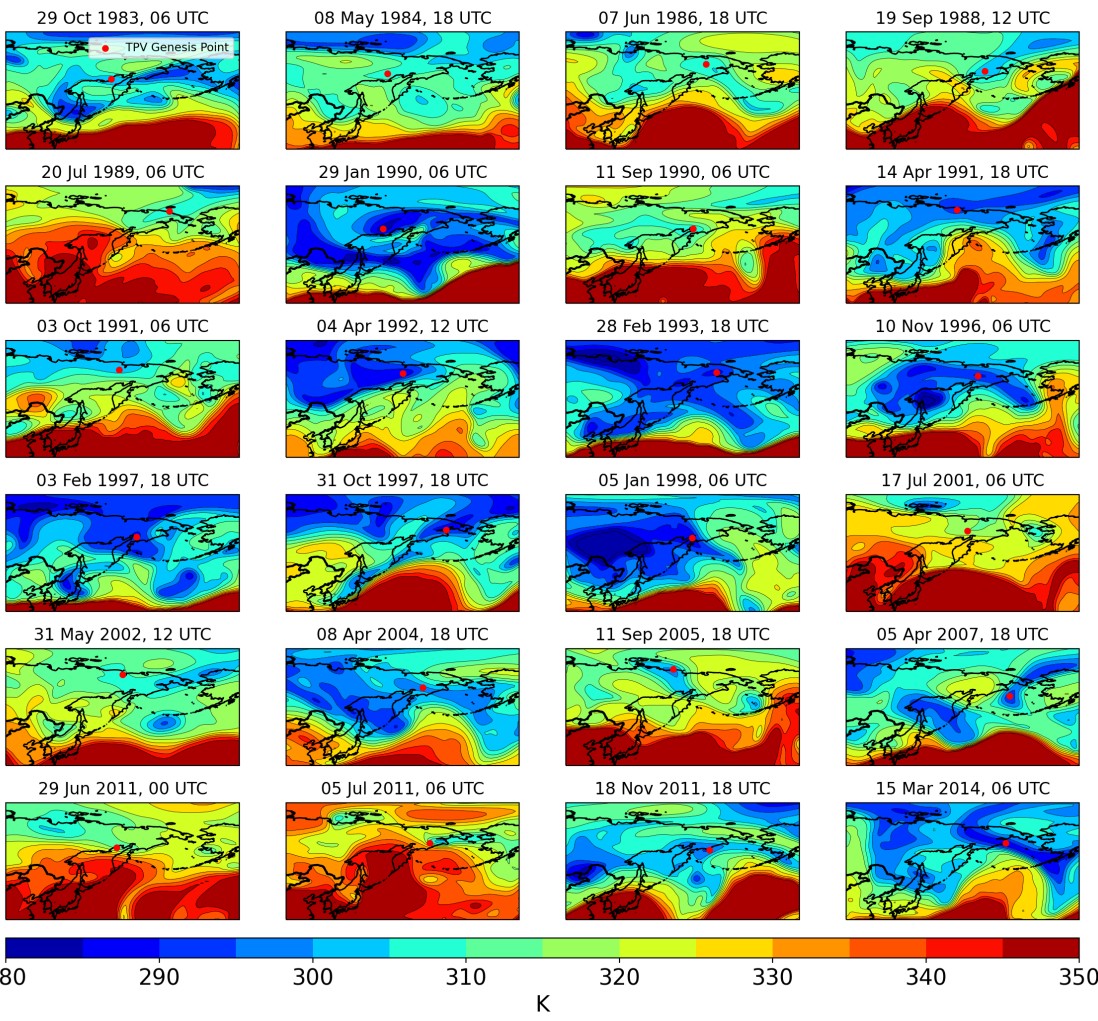

**Figure A3.** Maps of tropopause potential temperature (K; fill and contours) at the time of each TPV genesis event in the eastern Siberia non-likely split cluster. Red dots indicate the genesis location of the TPV.

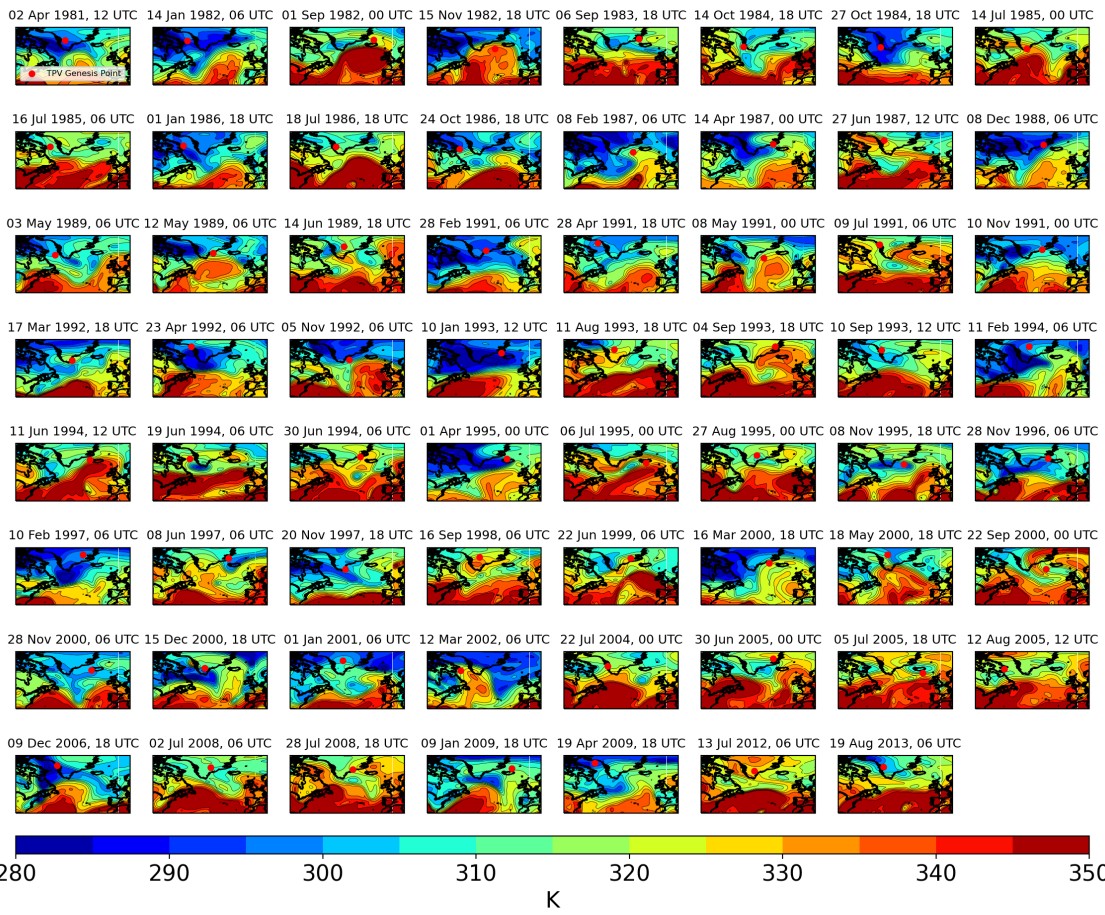

**Figure A4.** Maps of tropopause potential temperature (K; fill and contours) 24 hours prior to each TPV genesis event in Greenland non-likely split cluster. Red dots indicate the eventual genesis location of the TPV.

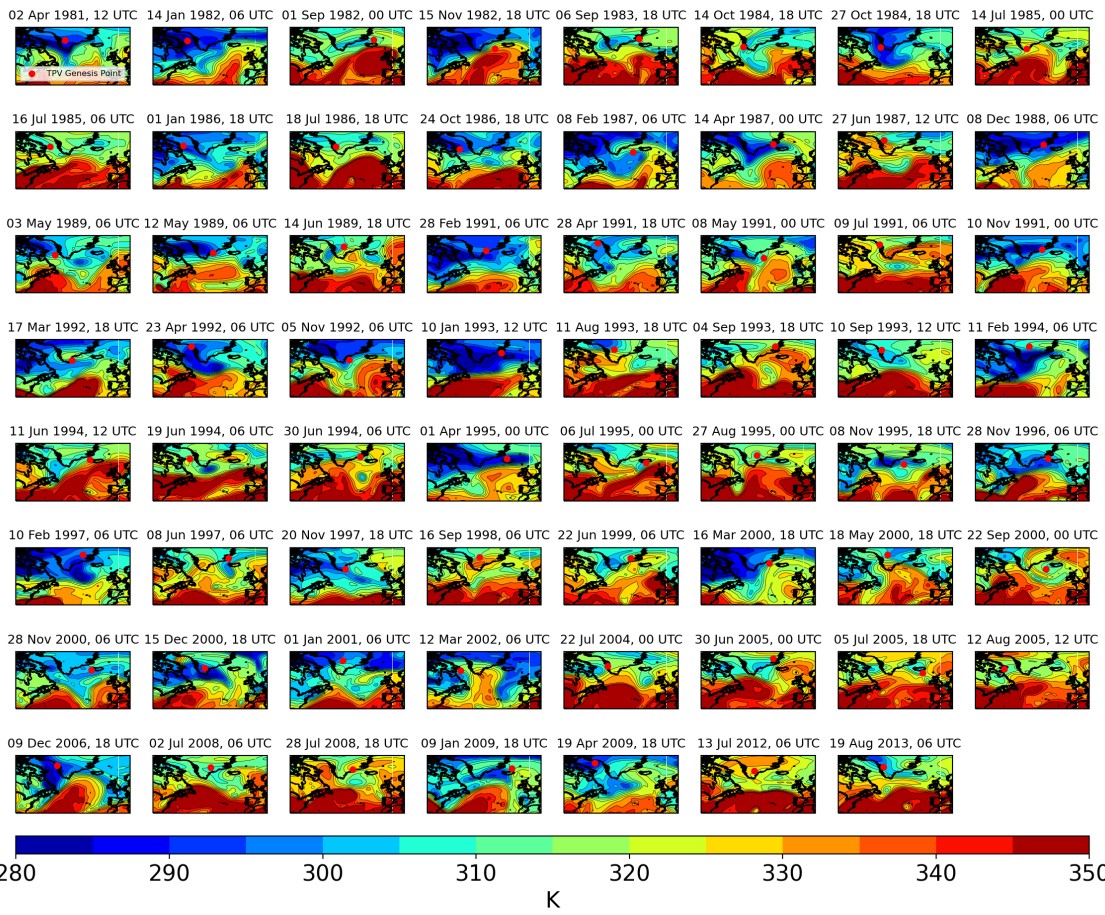

**Figure A5.** Maps of tropopause potential temperature (K; fill and contours) 12 hours prior to each TPV genesis event in the Greenland non-likely split cluster. Red dots indicate the eventual genesis location of the TPV.

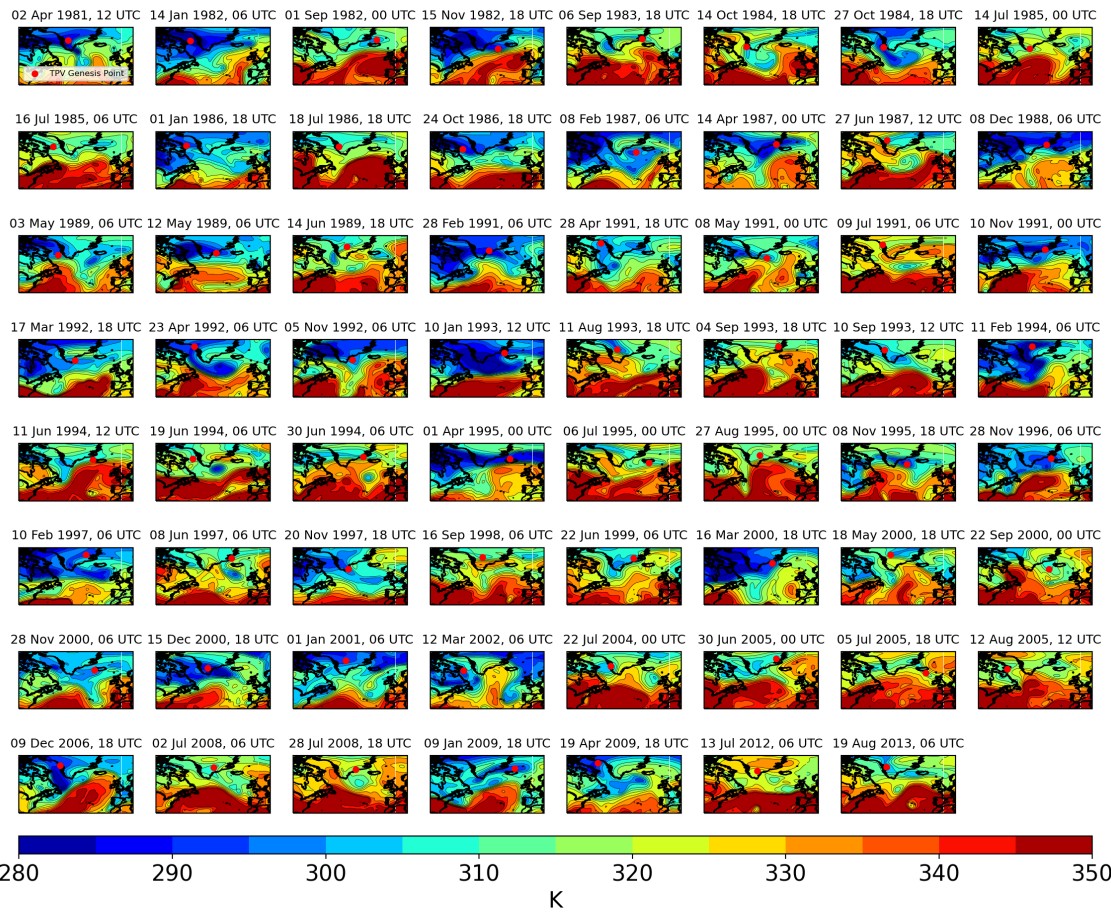

**Figure A6.** Maps of tropopause potential temperature (K; fill and contours) at the time of each TPV genesis event in the Greenland non-likely split cluster. Red dots indicate the genesis location of the TPV.

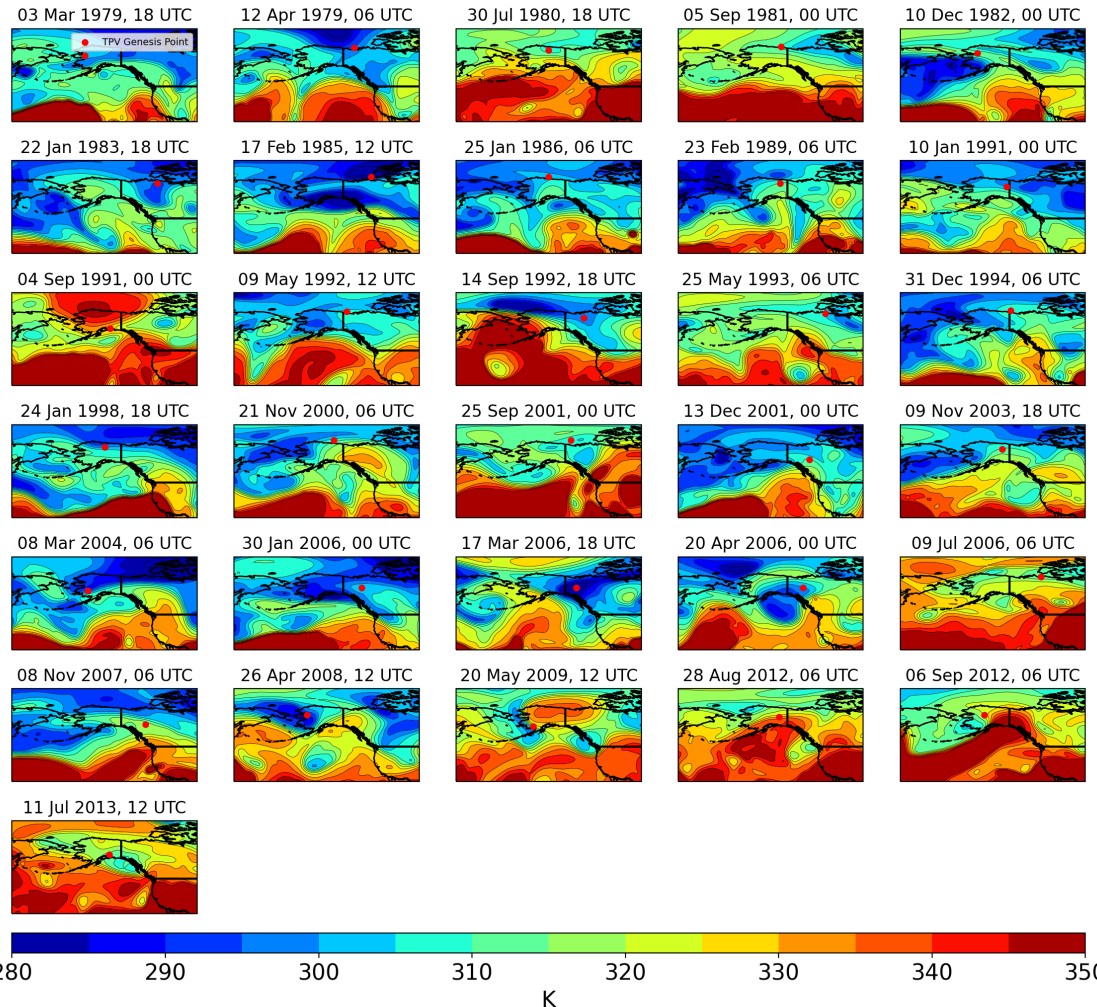

**Figure A7.** Maps of tropopause potential temperature (K; fill and contours) 24 hours prior to each TPV genesis event in the Alaska non-likely split cluster. Red dots indicate the eventual genesis location of the TPV.

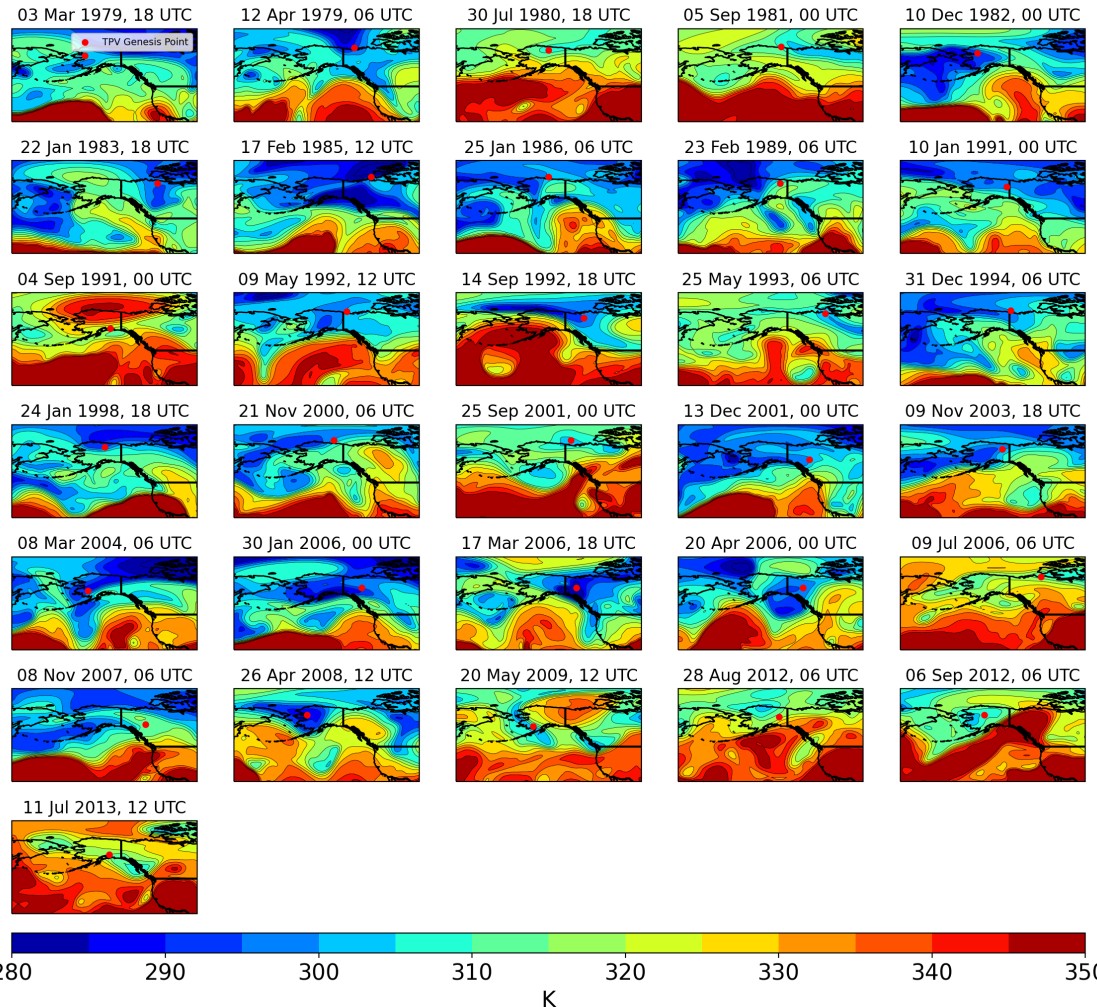

**Figure A8.** Maps of tropopause potential temperature (K; fill and contours) 12 hours prior to each TPV genesis event in the Alaska non-likely split cluster. Red dots indicate the eventual genesis location of the TPV.

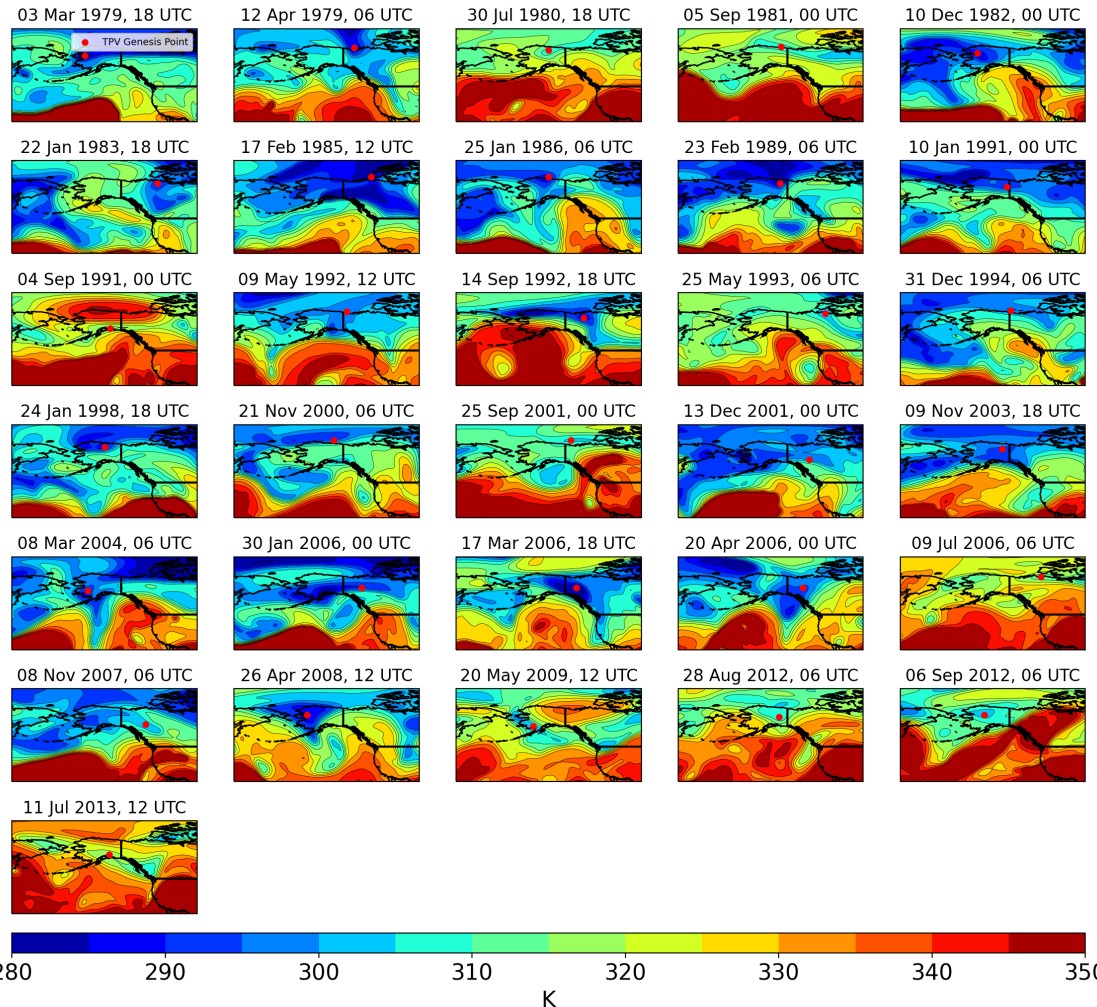

**Figure A9.** Maps of tropopause potential temperature (K; fill and contours) at the time of each TPV genesis event in the Alaska non-likely split cluster. Red dots indicate the genesis location of the TPV.