# Peer review of "Characteristics of long-track tropopause polar vortices"

_Weather and Climate Dynamics, 2021_

## Referee Comment (RC1)

Review of manuscript 2021 – 070 submitted to Weather and Climate Dynamics

**Characteristics of long-track tropopause polar vortices**

by M. T. Bray and S. M. Cavallo

Tropopause polar vortices (TPVs) are positive upper-level potential vorticity anomalies intensifying via longwave cooling in the Arctic. They are associated with severe weather events (e.g., cold air outbreaks) when moving out of the Arctic and they play an important role as a pre-cursor to surface cyclogenesis. Hence, understanding the processes driving their genesis and evolution is not only interesting from a scientific point of view but it is also highly relevant from a forecasting perspective. This study focuses on the most long-lived TPVs (i.e., longer than 2 weeks, corresponding to the 95[th] percentile in a TPV climatology) as unique cases for studying the conditions that favour TPV development. Specifically, the study first presents statistics of basic TPV characteristics such as intensity and size and then moves on to explore the processes leading to their genesis. While most of the TPVs form as splits from a pre-cursor TPV, characteristic patterns of Rossby wave breaking and ridge building are identified to favour TPV genesis in certain regions. Then, the evolution and tracks of these long-lived TPVs are explored. A particularly noteworthy result is the identification of two main pathways of TPVs to exit the Arctic, one via Canada and one via Siberia, as well as the observation that TPVs that exit the Arctic can live on for more than 10 days.

I found the study interesting to read and the results make for an important contribution towards a better understanding of TPVs. The methodology is sound, which applies in particular also to the thorough Monte-Carlo based statistical tests. In addition, the presentation is mostly clear and figures are of high quality. I have a few minor suggestions for improvement that I would like the authors to consider. Other than that, I am convinced that this study will make for a valuable contribution to WCD.

**General comments:**

- While the introduction gives a comprehensive exposition of the current knowledge about TPVs, there are some unnecessary jumps between topics and I suggest the authors re-arrange some of it. For example, the paragraph about impacts (L42ff) is squeezed in between two paragraphs that are concerned with the processes driving the formation and intensification of TPVs. Furthermore, genesis of TPVs and their subsequent intensification should be discussed in separate paragraphs. Finally, I think that the introduction could more strongly point out the gaps of knowledge that motivate studying long-lived TPVs.

- I find the diversity of synoptic configurations leading to the genesis of non-split TPVs fascinating and Figs. 4-6 (as well as those in the appendix) are very helpful for illustrating this diversity. However, I am not sure whether the way they are arranged now is ideal. I found myself switching back and forth several times between Figs. 4-6. Instead of arranging the panels chronologically by event with separate figures for different times relative to

genesis, I'd suggest to present events for each of the three categories in separate figures with panels corresponding to different relative times next to each other.

In addition, I would find it more intuitive if the schematic Fig. 7 was presented before the case studies. This would also help to streamline the section, the four typical patterns are explained twice – first in the context of the cases and second when explaining Fig. 7.

- An especially interesting result of this study is that TPVs that exit the Arctic can live on for many days and eventually move back to the Arctic, where they may re-intensify. The pathways sketched in Fig. 12 suggest that such TPVs travel along the main storm tracks in the North Atlantic and the North Pacific. Hence, I am wondering how unusual it is for a TPV to "survive" a passage through these regions? I imagine that a TPV approaching the main storm tracks will inevitably catch up with a baroclinic zone and then start to interact with it, i.e. trigger surface cyclogenesis. Could you say something about whether the long-lived TPVs are less often involved in surface cyclogenesis than ordinary TPVs and if so why? One way to approach this would be via TPV centred composites similar to Fig. 9 showing surface fields. Alternatively, also a matching of the TPV tracks with a cyclone data base could shed some light on this.

- I would have wished some more discussion of the results in the context of existing climatologies of stratospheric cutoffs and the processes that cause their genesis and demise (e.g., Portmann et al. 2021 and references therein), which are certainly related to the processes governing TPVs – at least once they have left the Arctic.

**Specific comments:**

(1) L12: The sentence beginning with "Notable differences emerge … " is a bit vague. Either remove or specifically state what the differences between long-lived TPVs in summer vs. winter are.

(2) Abstract: I'd find it worth to mention the main exit pathways from the Arctic of the long-lived TPVs as this results is directly relevant in terms of impact.

(3) L35: Start a new paragraph here since the formation of TPVs is a new topic.

(4) L58: Suggest to connect this paragraph to what is previously said about the dynamical processes and splitting of the vortices.

(5) L71 and also L375, L412: "low-shear" → "low shear" to be consistent with the rest of the paper

(6) L78: Suggest to remove the sentence "Thus, we hope…" or move it to the conclusion.

(7) L103: "the set … were created" → "the set … **was** created"

(8) L104: The reference to Dee et al. (2011) should be moved to the mention of the ERA-Interim in the previous sentence.

(9) L105: Are these always year-round climatologies or do they take the seasonal cycle into account?

(10) L132ff and Fig. 1: I undestand the distributions in (a) and (b) are per track. It is less clear to me in (c) – (f): are these distributions for all track points or only the extrema along the tracks? Please specify.

(11) L148: With "low shear", do you mean vertical or horizontal shear? To me vertical wind shear appears particularly detrimental for a TPV due to baroclinic interaction.

(12) L155: A high lysis frequency in the storm track regions is also found for stratospheric cutoffs (see Portmann et al. 2021).

(13) L162: No need to put the sentence "Note that when…" in brackets. I find this information relevant.

(14) L166: To what end? Do you mean to shed light on the split vs. non-split TPV genesis events?

(15) L203: How is the sorting of the non-likely split TPVs done? Is it based on visual inspection?

(16) L237: "method of genesis" sounds odd

(17) L275: I don't understand what you mean by "This feature of the composites is likely a reflection of mainly TPVs that form nearer to the mid-latitudes…"

(18) L270ff and Fig. 9: On what level are the winds? I assume it is the 2pvu tropopause. In any chase this should be specified.

(19) L317: What do you mean by "flow regime"?

(20) L322: "into two regions" → "**in** two regions"

(21) L326ff: This is a very interesting and – to me at least – surprising result. I would have expected that TPVs, once they exit the Arctic, will inevitably interact baroclinically with the lower troposphere and as a result decay.

(22) L421: "objective set **of** TPV genesis mechanism"

(23) L411: I suggest to streamline this last paragraph.

**References:**

Portmann, R., Sprenger, M., and Wernli, H.: The three-dimensional life cycles of potential vorticity cutoffs: a global and selected regional climatologies in ERA-Interim (1979–2018), Weather Clim. Dynam., 2, 507–534, https://doi.org/10.5194/wcd-2-507-2021, 2021.

---

## Referee Comment (RC2)

Review of wcd-2021-70:

"Characteristics of long-track tropopause polar vortices"
by
Matthew T. Bray and Steven M. Cavallo

**Recommendation: Minor revisions**

The manuscript described climatological aspects of long-lived TPVs using a TPV tracking algorithm. Certain geographic aspects are discussed, especially in terms of genesis of TPVs.
The manuscript is very well written and concise. The number of figures is rather on the limit, also given the complexity of some of the figures, see comments below.
Overall, the manuscript makes a valuable contribution to the field and could be recommended for publication after some comments below are addressed.

**General Comments:**

Regarding the TPVTrack algorithm, it is understandable that the authors mainly refer the reader to a previous paper outlining the method, but it would still be appreciated by the reader here is a brief description of potential thresholds used in the water-shedding are clarified. For example, if one continues to watershed, at some stage all objects will be unified into one object. The authors should briefly clarify what kind of thresholds are used in the water-shedding to better understand the identification of TPV objects on the PV2 surface. Also the equivalent radius at the end of section 2.1 is most likely depending on a threshold in the water-shedding that ultimately decides the size of the objects one would assume.

Given that there are so many clustering algorithms, it appears strange why the authors decided for a subjective classification given the large dataset that would certainly allow for some automatic clustering. Self-organizing-maps could have also been a rather useful method, given that one interest was the genesis environment. The authors should further substantiate the choice of a subjective clustering and in fact should consider automated schemes, such as self-organizing maps or similar to split the TPV tracks into different classes.

Given the track information of TPVs available to this study, it would have been of great value to include a discussion of the propagating nature of these systems. There is a discussion on genesis and general track density, as well as exit pathways for TPVs. However, to further substantiate the claim that these long-lived TPVs could contribute to enhancing predictability, it would be essential to know about their propagation characteristics and if these can be understood by general vortex or Rossby wave dynamics. The authors are strongly encouraged to consider expanding their analysis to address the kind of nature of propagation for TPVs in their database.

**Specific Comments:**
Reference to line numbers in the manuscript.

L17: The statement in the first sentence needs backing with literature. Alternatively, as evidence for the statement is provided in greater detail in the ensuing sentences, the first sentence might be removed.

L182: RWB

L220: The formulation "broad region of low potential temperature may itself be a large-scale mid-latitude PV anomaly or may include several embedded PV anomalies" should be clarified further. How could it be seen itself as a PV anomaly and what is meant by several embedded PV anomalies. The authors should more clearly separate between environment, anomalies, and even sub-anomalies if that is what they were aiming to refer to.

L348: "assuming they do so"? Why can the authors not be more certain about this given the track data they have?

L364-366: Please further clarify the tropopause folding all the way down to connect with the low-level stable layer. This seems rather extreme and would need further evidence or a citation presenting this claim.

Figures 4, 5, and 6 are difficult to digest given the wealth of information. The authors are encouraged to limit the number of panels to guide the reader more directly to the essential aspects that these figures are aiming to convey.

Similarly, Figures A1-6 would also benefit from a reduction in complexity. Potentially, the number of panels could be doable, though the type of shading should then be reconsidered to reduce complexity. The latter also applied to the comment on the figures in the main manuscript.

---

## Author Comment (AC1)

**Response to Reviewers: "Characteristics of long-track tropopause polar vortices"**

Original reviewer comments are in black, author responses are in red.

**General Notes:**

We would first like to thank the two anonymous reviewers for their detailed and helpful suggestions. We have used their comments to make improvements to the readability and completeness of the manuscript, as will be detailed below.

Significant changes include a reworking of the figures associated with non-likely split TPV genesis mechanisms (previously Fig. 4-6 and A1-A6). A selection of illustrative cases from each cluster for each genesis event type are now included in the main text to provide for smoother interpretation. The remaining cases have all been moved to the appendix so that they are available to interested readers.

Two new figures have also been added: one is a composite of surface MSLP and tropopause potential temperature (in response to a reviewer comment) which we use to characterize surface interactions throughout TPV lifetimes. The second shows results from a series of single column Rapid Radiative Transfer Model – Longwave (RRTM-LW) experiments which build off of Figure 8. While not specifically requested by reviews, this new figure reinforces the conclusions already drawn from Figure 8 with more quantitative backing (i.e., confirms what we had hypothesized previously). Thus, we feel that it is a useful addition to the manuscript and resolves many open questions about dynamic versus thermodynamic forcings.

**Reviewer 1**

While the introduction gives a comprehensive exposition of the current knowledge about TPVs, there are some unnecessary jumps between topics and I suggest the authors rearrange some of it. For example, the paragraph about impacts (L42ff) is squeezed in between two paragraphs that are concerned with the processes driving the formation and intensification of TPVs. Furthermore, genesis of TPVs and their subsequent intensification should be discussed in separate paragraphs. Finally, I think that the introduction could more strongly point out the gaps of knowledge that motivate studying long-lived TPVs.

We have adjusted the layout of the introduction. The impacts paragraph has been moved after the paragraphs about TPV formation and the genesis/RWB and intensification/dissipation paragraphs have been separated. We have also clarified the knowledge gaps in the final paragraph.

I find the diversity of synoptic configurations leading to the genesis of non-split TPVs fascinating and Figs. 4-6 (as well as those in the appendix) are very helpful for illustrating this diversity. However, I am not sure whether the way they are arranged now is ideal. I

found myself switching back and forth several times between Figs. 4-6. Instead of arranging the panels chronologically by event with separate figures for different times relative to genesis, I'd suggest to present events for each of the three categories in separate figures with panels corresponding to different relative times next to each other.

In addition, I would find it more intuitive if the schematic Fig. 7 was presented before the case studies. This would also help to streamline the section, the four typical patterns are explained twice – first in the context of the cases and second when explaining Fig. 7.

In response to this comment and similar comments from Reviewer 2, we've significantly changed the layout of these figures. As recommended, Figure 7 is now presented first. Figures 4-6 in the original manuscript have been moved to the Appendix along with their counterparts from the other two clusters. The subplots have been simplified (i.e., the amount of gradation has been reduced) in order to make reading these plots easier. For these appendix figures, we've kept the division by lag time in place. Having the panels chronologically ordered would result in either far too many subplots for the figure to be readable or require breaking the figures up at arbitrary points (even breaking the figures into type of genesis event would not work, as some of the clusters have too many CWB events to fit on one figure, for example). Since the appendix figures are only included for readers who are especially interested in examining the variety of synoptic patterns present, we've left this as is.

However, we have implemented the suggestion to order the subplots chronologically in the main body of the paper. For each of the three main genesis mechanisms (TPV splits are omitted), we have included a figure with an example case from each of the three clusters, with subplots in chronological order. That is, the new figure 5 will have a three-panel case of a cyclonic wave break event from the Siberia cluster, a three-panel case of a cyclonic wave break event from the Greenland cluster, and a three-panel case of a cyclonic wave break event from the Alaska cluster. Figures 6 and 7 will be the same but for anticyclonic wave breaks and ridge building cases. We believe that this will more simply illustrate different patterns for the most readers, without the need to comb through massive multi-panel plots.

An especially interesting result of this study is that TPVs that exit the Arctic can live on for many days and eventually move back to the Arctic, where they may re-intensify. The pathways sketched in Fig. 12 suggest that such TPVs travel along the main storm tracks in the North Atlantic and the North Pacific. Hence, I am wondering how unusual it is for a TPV to "survive" a passage through these regions? I imagine that a TPV approaching the main storm tracks will inevitably catch up with a baroclinic zone and then start to interact with it, i.e. trigger surface cyclogenesis. Could you say something about whether the long-lived TPVs are less often involved in surface cyclogenesis than ordinary TPVs and if so why? One way to approach this would be via TPV centred composites similar to Fig. 9 showing surface fields. Alternatively, also a matching of the TPV tracks with a cyclone data base could shed some light on this.

We've added a new figure as recommended which includes TPV centered composites with MSLP. It seems that long track TPVs are associated (on average) with a surface cyclone throughout their entire lifetime. At early stages, this is likely an Arctic cyclone developed from

interactions with the Arctic Frontal Zone, while near the end of the lifecycle, this may indicate interaction with a mid-latitude baroclinic zone. Though not included in the paper for brevity, similar composites of a sample of non-long-track TPVs are very similar, so it does not seem that the long-track TPVs are less likely to be involved with cyclogenesis. Rather, they are simply more resilient to the diabatic destruction effects that such an interaction can create.

To the point about how unusual it is for a TPV to survive a passage through the storm tracks, we've added numbers for how many TPVs reenter the Arctic into the text. Roughly 50% of all long-track TPVs that exit the Arctic and remain in the mid-latitudes for at least two days reenter the Arctic eventually. So, indeed, it seems that these TPVs entering the storm tracks and making it out is not all that uncommon.

I would have wished some more discussion of the results in the context of existing climatologies of stratospheric cutoffs and the processes that cause their genesis and demise (e.g., Portmann et al. 2021 and references therein), which are certainly related to the processes governing TPVs – at least once they have left the Arctic.

Thank you for suggesting the Portmann et al. paper, as it is very relevant to our study. We have added several references to findings from this study, especially those related to poleward of the jet PV cutoffs, which would largely overlap with TPVs that exit the Arctic. Some references from the Portmann paper that were not previously included here have been added for completeness.

Specific comments:

(1) L12: The sentence beginning with "Notable differences emerge … "is a bit vague. Either remove or specifically state what the differences between long-lived TPVs in summer vs. winter are.

Sentence clarified.

(2) Abstract: I'd find it worth to mention the main exit pathways from the Arctic of the longlived TPVs as this result is directly relevant in terms of impact.

Added.

(3) L35: Start a new paragraph here since the formation of TPVs is a new topic.

Paragraph break added.

(4) L58: Suggest to connect this paragraph to what is previously said about the dynamical processes and splitting of the vortices.

Transition added.

(5) L71 and also L375, L412: "low-shear" → "low shear" to be consistent with the rest of the paper

Done.

(6) L78: Suggest to remove the sentence "Thus, we hope…" or move it to the conclusion.

Removed.

(7) L103: "the set … were created" → "the set … was created"

Changed.

(8) L104: The reference to Dee et al. (2011) should be moved to the mention of the ERA-Interim in the previous sentence.

Done.

(9) L105: Are these always year-round climatologies or do they take the seasonal cycle into account?

The seasonal cycle is taken into account. Added.

(10) L132ff and Fig. 1: I undestand the distributions in (a) and (b) are per track. It is less clear to me in (c) – (f): are these distributions for all track points or only the extrema along the tracks? Please specify.

Included.

(11) L148: With "low shear", do you mean vertical or horizontal shear? To me vertical wind shear appears particularly detrimental for a TPV due to baroclinic interaction.

Vertical clear, this has been clarified in the text.

(12) L155: A high lysis frequency in the storm track regions is also found for stratospheric cutoffs (see Portmann et al. 2021).

A reference to this has been added in the text.

(13) L162: No need to put the sentence "Note that when…" in brackets. I find this information relevant.

Change made.

(14) L166: To what end? Do you mean to shed light on the split vs. non-split TPV genesis events?

Wording changed.

(15) L203: How is the sorting of the non-likely split TPVs done? Is it based on visual inspection?

Yes, clarified.

(16) L237: "method of genesis" sounds odd

Wording changed.

(17) L275: I don't understand what you mean by "This feature of the composites is likely a reflection of mainly TPVs that form nearer to the mid-latitudes…"

This was a poorly worded sentence and it has been changed significantly.

(18) L270ff and Fig. 9: On what level are the winds? I assume it is the 2pvu tropopause. In any chase this should be specified.

Indeed, tropopause winds; now specified.

(19) L317: What do you mean by "flow regime"?

We've changed this to background flow for clarity.

(20) L322: "into two regions" → "in two regions"

Change made.

(21) L326ff: This is a very interesting and – to me at least – surprising result. I would have expected that TPVs, once they exit the Arctic, will inevitably interact baroclinically with the lower troposphere and as a result decay.

See the earlier comment about cyclone composites, etc. This is indeed an interesting result. Our current hypothesis is that these TPVs have strengthened to a point that they are "resilient" enough to survive the baroclinic interaction before being slung back into the Arctic. As they are directly interacting with the surface, they likely decay, but not so much that they lose coherence.

(22) L421: "objective set of TPV genesis mechanism"

Correction made.

(23) L411: I suggest to streamline this last paragraph.

We've adjusted this paragraph some and tried to remove some of the excess wordiness.

**Reviewer 2**

Regarding the TPVTrack algorithm, it is understandable that the authors mainly refer the reader to a previous paper outlining the method, but it would still be appreciated by the reader here is a brief description of potential thresholds used in the water-shedding are clarified. For example, if one continues to watershed, at some stage all objects will be unified into one object. The authors should briefly clarify what kind of thresholds are used in the water-shedding to better understand the identification of TPV objects on the PV2 surface. Also the equivalent radius at the end of section 2.1 is most likely depending on a threshold in the water-shedding that ultimately decides the size of the objects one would assume.

This information (the relevant TPVTrack parameter is the filter distance, 300 km) has been added to the second paragraph of section 2a. This is the default filter distance for distinguishing unique extrema and watersheds for TPVTrack (and is reasonable since TPVs will tend to have

length scales around 300 km), which has been tested extensively for the ERA-I data that we use in this study.

Given that there are so many clustering algorithms, it appears strange why the authors decided for a subjective classification given the large dataset that would certainly allow for some automatic clustering. Self-organizing-maps could have also been a rather useful method, given that one interest was the genesis environment. The authors should further substantiate the choice of a subjective clustering and in fact should consider automated schemes, such as self-organizing maps or similar to split the TPV tracks into different classes.

With trends manually identified in this study, we do plan to use more algorithmic methods to study TPV genesis as a next step, including self-organizing maps. We have added text to the conclusion section to clarify this. We decided to do the manual classification for this study mainly to provide a solid starting point for future work. Very little at all was known going into this study about non-split genesis mechanisms, and we worried about using an automated procedure with no a priori knowledge of what patterns would be expected. The small case size allowed us to find the major categories by hand, which should help alleviate any concerns about whether the results from an automated method are physically meaningful, since we know roughly what kind of categories we should see.

On a more technical note, running an automated clustering algorithm will take a bit of creativity, based on some of our preliminary tests. Because of the variety of scales, orientations, etc. that these genesis events can occur in, grouping seemingly "like" cases in an automated manner may not be trivial. For instance, two cases that we identified as CWBs may differ in scale by several hundred kilometers and in trough axis tilt by 80°. Without a priori input, it's not a guarantee that an automatic algorithm would've clustered these two events, even though they are similar based on our physical understanding. In short, this is a very good suggestion, and will be one of our primary avenues for future work, perhaps meriting its own manuscript.

Given the track information of TPVs available to this study, it would have been of great value to include a discussion of the propagating nature of these systems. There is a discussion on genesis and general track density, as well as exit pathways for TPVs. However, to further substantiate the claim that these long-lived TPVs could contribute to enhancing predictability, it would be essential to know about their propagation characteristics and if these can be understood by general vortex or Rossby wave dynamics. The authors are strongly encouraged to consider expanding their analysis to address the kind of nature of propagation for TPVs in their database.

This is another excellent suggestion which we have found difficult to address thoroughly within the bounds of this study. The short answer (which we have added to the text, in the second to last paragraph of Section 3c) is that in the Arctic, the propagation characteristics are best understood by vortex/advection dynamics. In the midlatitudes, they are understood by Rossby wave dynamics. The issue is predicting when and why this transition between regimes occurred. We've attempted to find some pattern for this, either by compositing the ambient environment when the TPVs exit or looking for patterns in the TPV lifecycles as to when the Arctic exit

occurs. So far, this hasn't yielded any clear results. Below is a figure looking at a couple measures to look "wavy" behavior in the TPV movement. The non-linearity parameter should be <1 for a wave dominated regime and >1 for a vortex dominated regime. Early in the TPV lifetimes, this parameter is higher (indicating that the TPVs are in the Arctic and vortex dominated) while later on the parameter drops (as some TPVs have entered the mid-latitudes); however, there is no clear pattern as to when this happens, or common TPV characteristics when it occurs. Also included is a look at the average change in latitude over a 6-hour window (with the standard deviation of latitude changes shaded). We'd expect that in a wavier regime, TPV latitudes would fluctuate more significantly, and, indeed, the standard deviation envelope does increase near the end of the vortex lifetime when many of these TPVs would be exiting the Arctic. Again, though, there's no clear pattern as to when this "wavier" mid-latitude behavior will begin for any given TPV.

In addition, we've tried tracking potential vorticity gradients south of the TPV (as this would tend to be a good indicator of when the TPV is near the jet/more dominated by Rossby waves) and compositing/looking at EOFs of TPV exit environments. Both of these yielded messy results. In summary, vortex/advection dynamics and Rossby wave dynamics can both be used to understand TPV propagation depending on where the TPV is. Understanding what forces the transition between these two is beyond the scope of the current study.

[Figure]

Specific Comments:

Reference to line numbers in the manuscript.

L17: The statement in the first sentence needs backing with literature. Alternatively, as evidence for the statement is provided in greater detail in the ensuing sentences, the first sentence might be removed.

Added.

L182: RWB

Done.

L220: The formulation "broad region of low potential temperature may itself be a large-scale mid-latitude PV anomaly or may include several embedded PV anomalies" should be clarified further. How could it be seen itself as a PV anomaly and what is meant by several embedded PV anomalies. The authors should more clearly separate between environment, anomalies, and even sub-anomalies if that is what they were aiming to refer to.

Thank you for pointing this out. This was just wording from a previous version of the paragraph that got mixed into the new version, and the part about embedded PV anomalies has been removed. The multiple embedded PV anomalies are what appear after the broad region has been compressed.

L348: "assuming they do so"? Why can the authors not be more certain about this given the track data they have?

This was a poor wording choice and it has been revised. Only some TPVs that exit the Arctic will ever reenter; those that do reenter do so quickly.

L364-366: Please further clarify the tropopause folding all the way down to connect with the low-level stable layer. This seems rather extreme and would need further evidence or a citation presenting this claim.

We've clarified in the text that this hypothesis would only apply to some especially strong wintertime TPVs. We certainly have seen individual TPV cases which fold down to near the surface in the high Arctic, so the idea is possible in theory. As for how often it may happen, wintertime surface inversions over Arctic land are known to reach depths of at least 1 km or 150 hPa with regularity (Zhang et al. 2011, Zhang et al. 2021). For our long track TPVs, around 15-20% of winter TPVs involved a tropopause fold this deep at their point of maximum amplitude. So, for some cases, this hypothesis may indeed hold true. For other cases, the explanation may lie with the reduced shortwave radiation or reduced latent heating during the winter.

Figures 4, 5, and 6 are difficult to digest given the wealth of information. The authors are encouraged to limit the number of panels to guide the reader more directly to the essential aspects that these figures are aiming to convey.

Similarly, Figures A1-6 would also benefit from a reduction in complexity. Potentially, the number of panels could be doable, though the type of shading should then be reconsidered to reduce complexity. The latter also applied to the comment on the figures in the main manuscript.

As addressed in the response to reviewer 1, we have implemented these suggestions. Figures 4-6 have been replaced with single case studies for each cluster and for each genesis mechanism, making the figures much larger and easier to follow. The remaining plots have been added to the appendix. The postage stamp plots in the appendix have been greatly simplified (we reduced the number of shading levels by a factor of 10 and clipped the bounds of the colorbar to emphasize minima and maxima), which we believe greatly improves the readability of these figures.

**References**

Zhang, L., M. Ding, T. Dou, Y. Huang, J. Lv, and C. Xiao (2021): "The shallowing surface temperature inversions in the Arctic." *Journal of Climate,* **34** (10), 4159-4168.

Zhang, Y., D. Seidel, J. Golaz, C. Deser, and R. Tomas (2011): "Climatological characteristics of Arctic and Antarctic surface-based inversions." *Journal of Climate* **24** (19), 5167-5186.

---

## Author Response (AR2)

**Response to Reviewer, Technical Corrections: "Characteristics of long-track tropopause polar vortices"**

Original reviewer comments are in black, author responses are in red.

**Reviewer 1**

(1) RRTM-experiments: Could the authors say something about how the PV tendencies are computed from the radiative heating / cooling rates? Note that the PV tendency does not only depend on the vertical gradient of the heating rate, but it is also proportional to the absolute vorticity. Did you assume a typical value for the absolute vorticity when converting gradients of the heating rates to PV tendencies or did you take the actual vorticity profile? I think this should be specified in the paper. Also, I am wondering whether the cyclonic vorticity present (once the TPV exists) may help to further strengthen the PV rates?

Average vorticity profiles were taken and used in the same manner as the temperature and humidity profiles. This has been added to the text. To your second point, while we didn't explicitly explore this in the paper, we expect that that would be the case. As the TPV strengthens and additional cyclonic vorticity is present, the rate of PV development should increase as a result.

(2) L31: stratopshere → stratosphere

Corrected.

(3) L247: Do you mean a "shortwave trough"?

Shortwave ridge, added.

(4) L387: "...for winter case lysis events…" sounds odd, suggest rephrasing.

Rephrased.

(5) L416: predominate → predominant

Corrected.